# Memory Efficient Optimizers with 4-bit States

**Bingrui Li[1], Jianfei Chen[1†], Jun Zhu[1]**
[1]Dept. of Comp. Sci. and Tech., Institute for AI, BNRist Center, THBI Lab,
Tsinghua-Bosch Joint ML Center, Tsinghua University
lbr22@mails.tsinghua.edu.cn; {jianfeic, dcszj}@tsinghua.edu.cn

## Abstract

Optimizer states are a major source of memory consumption for training neural networks, limiting the maximum trainable model within given memory budget. Compressing the optimizer states from 32-bit floating points to lower bitwidth is promising to reduce the training memory footprint, while the current lowest achievable bitwidth is 8-bit. In this work, we push optimizer states bitwidth down to 4-bit through a detailed empirical analysis of first and second moments. Specifically, we find that moments have complicated outlier patterns, that current block-wise quantization cannot accurately approximate. We use a smaller block size and propose to utilize both row-wise and column-wise information for better quantization. We further identify a zero point problem of quantizing the second moment, and solve this problem with a linear quantizer that excludes the zero point. Our 4-bit optimizers are evaluated on a wide variety of benchmarks including natural language understanding, machine translation, image classification, and instruction tuning. On all the tasks our optimizers can achieve comparable accuracy with their full-precision counterparts, while enjoying better memory efficiency.[*]

## 1 Introduction

Large-scale models with a massive amount of parameters [5, 9, 20, 22, 49, 58] have shown impressive few-shot learning abilities on general tasks [52]. Despite being powerful, training these models is challenging. Memory capacity is one of the main bottlenecks of training large-scale models. Modern neural networks are typically trained with stateful optimizers such as Adam [26], which need to maintain one or two optimizer states (i.e., first and second moments) per each parameter. As the model size grows, the memory consumed by optimizer states can be a dominating factor of memory consumption [40, 41].

There are several attempts to reduce optimizers' memory consumption. Factorization [2, 8, 46] applies low-rank approximation to optimizer states, delta tuning [18, 23, 24, 27, 28] avoids maintaining optimizer states for most parameters by only tuning a small subset, and low-precision optimizers [15, 44] represent their states with low-precision numerical formats, which consume less memory.

Among these methods, low-precision optimizers are attractive due to their simplicity and wide applicability. Dettmers et al. [15] propose an 8-bit optimizer with reparameterized embedding layers ("stable embedding layers") and a block-wise 8-bit dynamic exponential numerical format for optimizer states. Their 8-bit optimizers achieve similar convergence to full-precision optimizers on language modeling, image classification, machine translation, and language understanding tasks.

In this work, we push the required numerical precision of low-precision optimizers from 8 to 4-bit through analyzing the patterns in the first and second moments and designing dedicated quantizers

---

[*]Code is available at https://github.com/thu-ml/low-bit-optimizers
[†]Corresponding author.

---

**Algorithm 1** Compression-based Memory Efficient Optimization Framework

---

**Require:** black-box stochastic optimization algorithm $\mathcal{A}$, initial parameter $\mathbf{w}_0 \in \mathbb{R}^p$, initial optimizer state $\bar{\mathbf{s}}_0 = \mathbf{0}$, total number of iterations $T$
 1: **for** $t = 1, 2, \ldots, T$ **do**
 2:     Sample a minibatch $\zeta_t$ and get stochastic gradient $\mathbf{g}_t = \nabla_{\mathbf{w}} f(\mathbf{w}_{t-1}, \zeta_t)$
 3:     $\mathbf{s}_{t-1} \leftarrow \text{decompress}(\bar{\mathbf{s}}_{t-1})$
 4:     $\mathbf{w}_t, \mathbf{s}_t \leftarrow \mathcal{A}(\mathbf{w}_{t-1}, \mathbf{s}_{t-1}, \mathbf{g}_t)$
 5:     $\bar{\mathbf{s}}_t \leftarrow \text{compress}(\mathbf{s}_t)$
 6: **end for**
 7: **return** $\mathbf{w}_T$

---

for optimizer states. Specifically, we find that moments exhibit complicated outlier patterns, that vary across different parameter tensors. The large block size proposed by Dettmers et al. [15] cannot properly handle all different outlier patterns. Based on this observation, we propose to use a smaller block size, which improves the approximation of first moment.

For the second moment, we find its quantization suffers from a *zero-point problem*. Since the update direction is usually inversely proportional to the square root of the second moment, quantizing non-zero quantities to zero will cause significant deviation. To address this problem, we propose a simple linear quantizer to exclude the zero-point for second moment. We further propose a stronger quantization technique, *rank-1 normalization*, to improve the approximation of second moment by better handling the outlier patterns. Our proposed quantization techniques are robust enough to achieve lossless convergence under 4-bit, even without the stable embedding layers proposed by Dettmers et al. [15].

Finally, we investigate the combination of factorization methods [2, 46] with low-precision optimizers, and propose a memory efficient optimizer which utilizes quantized 4-bit first moment and factorized second moment. For applicable tasks, the hybrid optimizer enjoys best of both worlds: good convergence and memory efficiency.

We evaluate our 4-bit optimizers on a wide range of tasks, including natural language understanding, machine translation, image classification, and instruction tuning of large language models. On all the benchmarks, our 4-bit optimizer can converge similarly fast with full-precision optimizers, and the converged models do not have noticeable accuracy loss. Our optimizers consumes less memory than existing 8-bit optimizers [15], while improves the throughput of language model finetuning with optimizer offloading [41, 45] due to reduced communication cost.

## 2 Preliminaries

In this section, we present some preliminaries of compression-based memory efficient optimizers and discuss quantization methods for compression of optimizer states in a general formulation.

### 2.1 A Framework for Compression-based Memory Efficient Optimizers

Gradient-based stateful optimizers like SGDM [38, 47], Adam [26], AdamW [32] are the principal choices in deep neural network training. However, the memory footprint of stateful optimizers is several times of model itself, which results in a bottleneck for large model pretraining/finetuning. Consider the update rule of the Adam optimizer:

$$\mathbf{Adam}(\mathbf{w}_{t-1}, \mathbf{m}_{t-1}, \mathbf{v}_{t-1}, \mathbf{g}_t) = \begin{cases} \mathbf{m}_t & = \beta_1 \mathbf{m}_{t-1} + (1 - \beta_1)\mathbf{g}_t \\ \mathbf{v}_t & = \beta_2 \mathbf{v}_{t-1} + (1 - \beta_2)\mathbf{g}_t^2 \\ \hat{\mathbf{m}}_t & = \mathbf{m}_t/(1 - \beta_1^t) \\ \hat{\mathbf{v}}_t & = \mathbf{v}_t/(1 - \beta_2^t) \\ \mathbf{w}_t & = \mathbf{w}_{t-1} - \eta \cdot \hat{\mathbf{m}}_t / (\sqrt{\hat{\mathbf{v}}_t} + \epsilon) \end{cases} \tag{1}$$

During the training process, the model parameters $\mathbf{w}_t$ and optimizer states (i.e., first and second moments) $\mathbf{m}_t$, $\mathbf{v}_t$ need to be stored persistently in the GPU memory. As the model grows large, optimizer states are a main source of training memory consumption. Each parameter dimension takes

two 32-bit optimizer states, making the memory consumption of Adam-style optimizers three times larger than stateless optimizers like SGD.

Compressing the optimizer states is a promising method to achieve memory efficient optimization. Formally, given a gradient-based optimizer $\mathcal{A}$, a memory efficient version of the optimization algorithm with compressed optimizer states is given by Alg. 1. The algorithm $\mathcal{A}$ can be any gradient-based optimizers like SGDM, Adam, AdamW, etc. See App. F for examples. In Alg. 1, the precise states $\mathbf{s}_t$ are only temporal, and only compressed states $\bar{\mathbf{s}}_t$ are stored persistently in the GPU memory. Memory footprint reduction can be achieved since in neural networks, the state vectors $\mathbf{m}_t$ and $\mathbf{v}_t$ are usually concatenation of state vectors of each parameterized layer. Therefore, we can perform the optimizer steps (Line 3-5) separately for each layer, so only one single layer of the precise states are presented in the memory at a time. The states for all other layers are kept compressed. In the rest of this paper, we focus on the compression and decompression method, to achieve high compression rate of optimizer states while maintaining good convergence of the optimizer.

## 2.2 Main Compression Method: Quantization

Quantizing the optimizer states to lower precision (e.g. 8-bit integers) is an effective way to compression optimizer states [15]. In this case, the optimizer states are compressed with a quantizer and decompressed with a dequantizer. The low-precision numerical format significantly impacts the accuracy of quantization methods. Here, we present a general framework for numerical formats we considered for compressing optimizer states.

A quantizer converts full-precision tensors to low-precision formats. Based on the formulation proposed by Dettmers and Zettlemoyer [16], we disentangle the quantizer $\mathbf{Q}(\cdot)$ into two parts: normalization $\mathbf{N}(\cdot)$ and mapping $\mathbf{M}(\cdot)$, which applies sequentially and element-wisely to a tensor to be quantized. For concise presentation, we only discuss quantizers for unsigned inputs, and defer the discussion of signed inputs in App. E.1. Formally, the quantizer for a tensor $\mathbf{x} \in \mathbb{R}^p$ is given by

$$q_j := \mathbf{Q}(x_j) = \mathbf{M} \circ \mathbf{N}(x_j).$$

**Normalization**    The normalization operator $\mathbf{N}$ scales each elements of $\mathbf{x}$ into the unit interval, i.e. $[0, 1]$. Normalization can have different granularity, such as per-tensor, per-token (row) [37, 55], per-channel (column) [4], group-wise [37, 55] and block-wise [15]. The per-tensor and block-wise normalization operators are given by

$$n_j := \mathbf{N}_{\text{per-tensor}}(x_j) = x_j / \max_{1 \leq i \leq p} |x_i|,$$

$$n_j := \mathbf{N}_{\text{block-wise}}(x_j) = x_j / \max \left\{ |x_i| : 1 + B \lfloor j/B \rfloor \leq i \leq B \left( \lfloor j/B \rfloor + 1 \right) \right\},$$

respectively, where the involved scaling factors are called *quantization scale*, which are persistently stored together with quantized tensor until dequantization. The granularity of normalization presents a trade-off of quantization error and memory overhead. Normalization method with low quantization error and acceptable memory overhead is preferred. In this case, the coarsest per-tensor normalization operator has negligible memory overhead, i.e. only 1 scaling factor regardless of tensor size, but suffers from largest error due to outliers. Block-wise normalization views the tensor as an 1-dimensional array, divides the array into blocks of size $B$ called block and assigns a quantization scale within each block, which leads to $\lceil p/B \rceil$ quantizaton scales totally. The block size could be adapted to trade-off quantization error and memory overhead.

**Mapping**    A mapping [15] converts normalized quantities to low-bitwidth integers. Formally, the mapping operator $\mathbf{M} = \mathbf{M}_{\mathbf{T},b}$ is equipped with a bitwidth $b$ and a predefined increasing mapping, named *quantization mapping* $\mathbf{T} : [0, 2^b - 1] \cap \mathbb{Z} \to [0, 1]$. Then $\mathbf{M}$ is defined as

$$q_j := \mathbf{M}(n_j) = \arg \min_{0 \leq i < 2^b} |n_j - \mathbf{T}(i)|.$$

The design of $\mathbf{T}$ is critical as it could effectively mitigate quantization error by capturing the distribution information of $\mathbf{n}$. There are two kinds mappings that are of specific interest to optimizer states quantization, linear mapping and dynamic exponent (DE) mapping [13]. A linear mapping $\mathbf{T}(i) = (i + 1)/2^b$ defines a linear quantizer, where the quantization intevals distribute evenly in each block. The DE mapping can approximate small values well, similar to floating point numbers. DE splits the binary representation of a low-precision integer $i$ into three parts: a leading

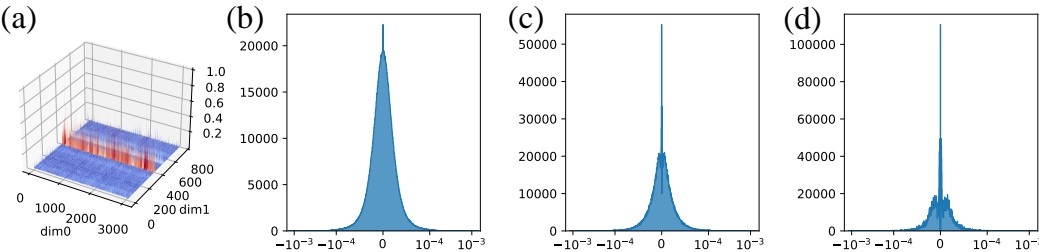

Figure 1: Visualization of the first moment in the `layers.3.blocks.1.mlp.fc1` layer in a Swin-T model. (a): Magnitude of the first moment. (b): Histogram of the first moment. (c): Moment approximated by `B128/DE`. (d): Moment approximated by `B2048/DE`.

sequence of $E$ zeros, followed by an indicator bit one, and remaining $F$ fraction bits. DE defines $\mathbf{T}(i) = 10^{-E(i)}\text{fraction}(i)$, where $\text{fraction}(i) \in [0.1, 1]$. See App. E.2 for full specifications and visualizations of different quantization mappings.

The normalization $\mathbf{N}$ and mapping $\mathbf{M}$ roughly play the same role in finding good quantization point candidates and they affect each other. If an oracle normalization scaling the original tensor $\mathbf{x}$ to a uniform distribution is accessible, linear mapping could be used generally. On the contrary, if the optimal mapping could be readily identified with respect to certain metrics for a per-tensor normalized tensor, there is no necessity to develop additional normalization methods that incur extra overhead for mitigating the negative effects of outliers. In essence, if one of the two operators approaches optimality, the quantizer can still perform well even when the other operator is set to its most basic configuration. Additionally, as one operator becomes increasingly powerful, the potential for improvement in the other operator gradually diminishes.

**Dequantization**   The dequantizer is just the inverse of the quantizer, which is simply

$$\tilde{x}_j := \mathbf{N}^{-1} \circ \mathbf{T}(q_j).$$

Based on our formulation, we name quantizers by their normalization and mapping methods as `Norm./Map.`. For example, 8-bit optimizers [15] use block-wise normalization with a block size of 2048 and dynamic exponent mapping, which we call `B2048/DE` in the rest of the paper.

## 3   Compressing First Moment

In the next two sections, we describe our design of compression and decompression methods in Alg. 1 to realize our memory efficient 4-bit optimizers. We first discuss the compression of the first moment. Our compression method for first moment is based on Dettmers et al. [15], which uses block-wise normalization with a block size of 2048 and dynamic exponent mapping [13]. We preliminary reduce the bitwidth from 8-bit to 4-bit and discover that the first moment is rather robust to quantization. This simple optimizer can already converge with 4-bit first moment, though in some cases there is accuracy degradation.

To further improve the performance, we investigate the patterns of the first moment. There are some outliers in the moments and outliers significantly affect quantization scale due to their large magnitude. Outlier patterns have been studied for weights and activations. It is shown that the weights are rather smooth [14, 54, 57], while the activation have column-wise outliers [4, 14, 53, 54], i.e., the outliers in activation always lie in some fixed channels. However, we find that the outlier patterns in optimizer states are quite complicated. Specifically, Fig. 2 shows patterns of first moment in a Swin-T model during training. Outliers of the `layers.3.blocks.0.mlp.fc1` layer lie in fixed rows while outliers of the `layers.3.blocks.0.mlp.fc2` layer lie in fixed columns. Actually, the outlier patterns vary across different architectures, layers and parameters. See more patterns in App. B.

The complicated outlier pattern makes optimizer states harder to quantize. There exist some moment tensors, that the outliers persist roughly in certain columns (dimension 1), as shown in Fig. 1. Block-wise normalization treats this tensor as an flattened one-dimensional sequence, in the row-first order. Therefore, any block size of 2048 would include an entry in the outlier column, resulting

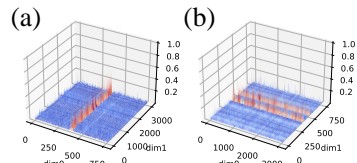

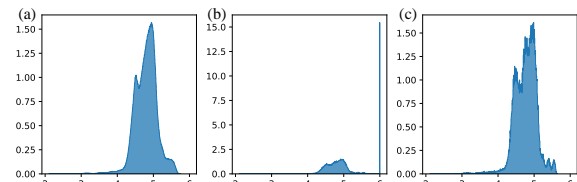

Figure 2: Outlier patterns vary across two first moment tensors. (a): outliers lie in fixed rows (dimension 0). (b): outliers lie in fixed columns (dimension 1).

Figure 3: Histogram of the inverse square root of second moment. (a) full-precision; (b) quantized with `B128/DE`; (c) quantized with `B128/DE-0`. All figures are at log10 scale and y-axis represents density.

in a large quantization scale. In this case, block-wise normalization is not better than per-tensor normalization. Therefore, we adopt a smaller block size of 128, as it provides enhanced performance while incurring only a little memory overhead. In Fig. 1, we show that quantizing with a smaller block size approximates the moments better than a large block size.

# 4 Compressing Second Moment

Compared to the first moment, quantizing the second moment is more difficult and incurs training instability. Besides its sharper outlier pattern and ill-conditioned distribution compared with first moment, we further identify the *zero-point problem* as the main bottleneck of quantizing the second moment. We propose an improved normalization method with a quantization mapping excluding zero. We also propose a factorization method for compressing the second moment, which leads to even better memory efficiency.

## 4.1 Zero-point Problem

The problem of quantizing second moment different from quantizing other tensors in neural networks. For weights [25], activations [25], and gradients [1], it is desirable to contain zero in the quantization mapping for lower approximation error and training stability. Empirically, zero is often the most frequent element [57]. But for second moment in Adam, small values around zero significantly impact the update direction, which is proportional to the inverse square root of the second moment, as shown in Eq. 1. In this case, a small quantization error of values around zero will cause catastrophically large deviation of the update direction. Fig. 3 shows the histogram of the inverse square root (i.e., transformed with $h(v) = 1/\left(\sqrt{v} + 10^{-6}\right)$) of the second moment quantized with `B128/DE`. The quantizer pushes most of entries of the tensor to zero, so the inverse square root of most points fall into $10^6$ due to zero-point problem, resulting in a complete degradation in approximation. One remedy is to simply remove zero from the DE quantization map, which we call `DE-0`. The smallest number representable by `DE-0` is 0.0033. With the `B2048/DE-0` quantizer applied to second moment, the approximation of second moment is more precise (Fig. 3), and the training with 4-bit optimizer stabilizes (Tab. 1). However, by removing zero, `DE-0` wastes one of the $2^4 = 16$ quantization points. We propose to use a `Linear` mapping $\mathbf{T}(i) = (i+1)/2^b$, whose smallest representable number is 0.0625. The linear mapping performs better than `DE-0` for quantizing second moment.

In Tab. 1, we ablate different quantization schemes and show that excluding zero from the mapping is indeed the key factor for second moment quantization, which cannot be replaced by a smaller block size or/and stochastic rounding [11]. We also test Stable Embedding layers proposed by Dettmers et al. [15], which are reparameterized embedding layers which can be more stably optimized. While Stable Embedding could improve training stability, it cannot fully retain accuracy since non-embedding layers still suffer from zero points. On the other hand, when the zero-point problem is properly addressed, Stable Embedding is no longer required to retain accuracy.

## 4.2 Rank-1 Normalization

We propose an empirically stronger normalization method named *rank-1 normalization* based on the observation of the heterogeneous outlier pattern in Sec. 3, inspired by the SM3 optimizer [2]. Formally, for a matrix-shaped non-negative (second moment) tensor $\mathbf{x} \in \mathbb{R}^{n \times m}$, its 1-dimensional

Table 1: Ablation analysis of 4-bit optimizers on the second moment on the GPT-2 Medium E2E-NLG finetuning task. The first line barely turns 8-bit Adam [15] into 4-bit, i.e. B2048/DE for both first and second moments. We only vary the quantization scheme for second moment. SR=stochastic rounding (see App. E.3 for details). Stable Embedding layers are not quantized. 32-bit AdamW achieves a BLEU of 67.7.

| Normalization | Mapping | Stable Embed.* | Factorized | Unstable(%) | BLEU |
|---|---|---|---|---|---|
| B2048 | DE | ✗ | ✗ | 33 | $66.6 \pm 0.61$ |
| B2048 | DE | ✓ | ✗ | 0 | $66.9 \pm 0.52$ |
| B128 | DE | ✗ | ✗ | 66 | $65.7 \pm$ N/A |
| B128 | DE+SR* | ✗ | ✗ | 33 | $65.4 \pm 0.02$ |
| B128 | DE | ✓ | ✗ | 0 | $67.2 \pm 1.13$ |
| B2048 | DE-0 | ✗ | ✗ | 0 | $67.5 \pm 0.97$ |
| B2048 | DE-0 | ✓ | ✗ | 0 | $67.1 \pm 1.02$ |
| B128 | DE-0 | ✗ | ✗ | 0 | $67.4 \pm 0.59$ |
| Rank-1 | DE-0 | ✗ | ✗ | 0 | $67.5 \pm 0.58$ |
| Rank-1 | Linear | ✗ | ✗ | 0 | $\mathbf{67.8 \pm 0.51}$ |
| Rank-1 | Linear | ✗ | ✓ | 0 | $\mathbf{67.6 \pm 0.33}$ |

statistics $\mathbf{r} \in \mathbb{R}^n$ and $\mathbf{c} \in \mathbb{R}^m$ are defined as $r_i = \max_{1 \le j \le m} x_{i,j}$ and $c_j = \max_{1 \le i \le n} x_{i,j}$, which are exactly the quantization scales of per-row and per-column normalization. Rank-1 normalization utilizes two scales jointly and produce a tighter bound for entry, which is defined as

$$\mathbf{N}_{\text{rank-1}}(x_{i,j}) = \frac{1}{\min\{r_i, c_j\}} x_{i,j}.$$

Rank-1 normalization could be easily generalized to high-dimensional tensors and signed tensors. See App. G for details and the pseudocode.

Compared with per-tensor, per-token (row), and per-channel (column) normalization, rank-1 normalization utilizes the 1-dimensional information in a more fine-grained manner and gives element-wisely unique quantizaion scales. It deals with the outliers more smartly and effectively when outlier persists in fixed rows or columns but the pattern across tensors are unknown and/or varied (Fig. 2). On the other hand, block-wise normalization is also capable to capture the local information and avoid outliers effectively regardless of the patterns when a small block size is taken, but rank-1 normalization provides a better trade-off between memory overhead and approximation. Rank-1 normalization falls back to per-tensor normalization for 1-dimensional tensors, so we use B128 normalization in those cases. Empirically, rank-1 normalization match or exceed the performance of B128 normalization at a moderate model scale (hidden_size=1024). It is also possible to combine block-wise and rank-1 normalization together, which we leave for future work.

### 4.3 Factorization

Many memory efficient optimization methods [2, 8, 46] represent the entire second moment with a few number of statistics, which differ from quantization and they take only sublinear memory cost. These works bring more memory saving but they are only applicable to the second moment. In this work, we find the factorization method proposed in Adafactor [46] could also avoid zero-point problem effectively, as shown in Tab. 1. Further, we explore the effects of factorization on second moment based on quantized optimizers to attain maximal memory saving while maintain lossless accuracy. Specifically, when factorization is enabled, we factorize all second moment with dimension greater than 1, and quantize 1d second moment tensors.

## 5 Experiments

We compare our 4-bit optimizers with their full-precision counterparts, as well as other memory efficient optimizers including 8-bit AdamW [15][†], Adafactor [46] and SM3 [2]. 8-bit AdamW's optimizer states are not quantized for embedding layers. For Adafactor, we compare both the

---

[†] https://github.com/TimDettmers/bitsandbytes

Table 2: Performance on language and vision tasks. Metric: NLU=Mean Accuracy/Correlation. CLS=Accuracy. NLG=BLEU. QA=F1. MT=SacreBleu. $^{\dagger}$: do not quantize optimizer states for embedding layers; $^{\ddagger}$: $\beta_1 = 0$. See more results in App. A.

| Optimizer | NLU RoBERTa-L | CLS Swin-T | NLG GPT-2 M | QA RoBERTa-L | MT Transformer |
|---|---|---|---|---|---|
| 32-bit AdamW | $88.9 \pm 0.01$ | $81.2 \pm 0.05$ | $67.7 \pm 0.67$ | $94.6 \pm 0.13$ | $26.61 \pm 0.08$ |
| 32-bit Adafactor | $89.1 \pm 0.00$ | $80.0 \pm 0.03$ | $67.2 \pm 0.81$ | $94.6 \pm 0.14$ | $26.52 \pm 0.02$ |
| 32-bit Adafactor$^{\ddagger}$ | $89.3 \pm 0.00$ | $79.5 \pm 0.05$ | $67.2 \pm 0.63$ | $94.7 \pm 0.10$ | $26.45 \pm 0.16$ |
| 32-bit SM3 | $87.5 \pm 0.00$ | $79.0 \pm 0.03$ | $66.9 \pm 0.58$ | $91.7 \pm 0.29$ | $22.72 \pm 0.09$ |
| 8-bit AdamW$^{\dagger}$ | $89.1 \pm 0.00$ | $81.0 \pm 0.01$ | $67.5 \pm 0.87$ | $94.5 \pm 0.04$ | $26.66 \pm 0.10$ |
| 4-bit AdamW (ours) | $89.1 \pm 0.01$ | $80.8 \pm 0.02$ | $67.8 \pm 0.51$ | $94.5 \pm 0.10$ | $26.28 \pm 0.05$ |
| 4-bit Factor (ours) | $88.9 \pm 0.00$ | $80.9 \pm 0.06$ | $67.6 \pm 0.33$ | $94.6 \pm 0.20$ | $26.45 \pm 0.05$ |

$\beta_1 > 0$ and the $\beta_1 = 0$ (no first moment) configuration. For our 4-bit optimizers, we report two versions both based on 32-bit AdamW: (1) "4-bit AdamW" quantizes first moment with `B128/DE` and second moment with `Rank-1/Linear`. (2) the more memory efficient "4-bit Factor" quantizes first moment with `B128/DE`, factorizes second moment when the dimension of tensor is greater than 1, and quantizes lefted 1-dimensional second moment with `Rank-1/Linear`. See App. D for details about the experiments.

**Models, datasets and hyperparameters**  We report performance metrics on standard benchmarks, including image classification (CLS) with Swin-T [31]$^{\ddagger}$ on ImageNet-1k [12], natural language understanding (NLU) by fine-tuning RoBERTa-L [30]$^{\S}$ fine-tuning on GLUE [51], question answering (QA) by fine-tuning RoBERTa-L on SQuAD [42, 43], natural language generation (NLG) by fine-tuning GPT-2 Medium [39]$^{\P}$ on E2E-NLG [35], machine translation (MT) by training Transformer-Base [50]$^{\|}$ on WMT14 en-de [3] and LLaMA [49] fine-tuning. We fine-tune LLaMA-7B, LLaMA-13B and LLaMA-33B [49] on the Alpaca dataset [48]$^{**}$ and evaluate them on MMLU [21] and standard common sense reasoning benchmarks: HellaSwag [56], ARC easy and challenge [10] and OpenBookQA [33].

We mainly follow the hyperparameters in the original paper or/and codebase. In each benchmark, we keep same hyperparameters in one optimizer on different quantization schemes, which gives an out-of-box transfer from full-precision optimizer to low-bit optimizer without extra hyperparameter tuning. See App. D for hyperparameters and training details.

**Accuracy of 4-bit Optimizers**  We first check whether our memory efficient 4-bit optimizers could retain accuracy. According to Tab. 2, our 4-bit optimizers can match or exceed 32-bit AdamW performance on all fine-tuning tasks (NLU, QA, and NLG) and are comparable on all pretraining tasks (CLS and MT). Sublinear memory optimizers Adafactor ($\beta_1 = 0$) and SM3 could have better memory efficiency, but they suffer from performance degradation, particularly on the CLS task. According to Tab. 3, our 4-bit AdamW will not destroy the capability of pretrained models while enabling them to obtain instruction-following ability. 4-bit AdamW is comparable with 32-bit AdamW on all tasks and does not get worse when the model size grows. Moreover, their convergence curves closely align (Fig. 4).

**Memory and Computing Efficiency**  We evaluate the memory and computation efficiency of proposed 4-bit optimizers on instruction tuning, NLU, and NLG tasks, in Tab. 4. Our 4-bit optimizer offers more memory saving compared to 8-bit optimizers, reducing the training memory consumption by up to 57.7%. It may look like the memory saving saturates when the bitwidth goes down. This is because we report the total memory consumption (including data, activations, and memory fragments)

---

$^{\ddagger}$`https://github.com/microsoft/Swin-Transformer`

$^{\S}$`https://github.com/huggingface/transformers`

$^{\P}$`https://github.com/microsoft/LoRA`

$^{\|}$`https://github.com/NVIDIA/DeepLearningExamples/tree/master/PyTorch/Translation/Transformer`

$^{**}$`https://github.com/tatsu-lab/stanford_alpaca`

Table 3: Performance on LLaMA fine-tuning on MMLU and commonsense reasoning tasks across different sizes.

| Model | Optimizer | MMLU (5-shot) | HellaSwag | ARC-e | ARC-c | OBQA |
|-------|-----------|---------------|-----------|-------|-------|------|
| LLaMA-7B | Original | 33.1 | 73.0 | 52.4 | 40.9 | 42.4 |
| | 32-bit AdamW | 38.7 | 74.6 | 61.5 | 45.1 | 43.4 |
| | 4-bit AdamW | 38.9 | 74.7 | 61.2 | 44.4 | 43.0 |
| LLaMA-13B | Original | 47.4 | 76.2 | 59.8 | 44.5 | 42.0 |
| | 32-bit AdamW | 46.5 | 78.8 | 63.6 | 48.3 | 45.2 |
| | 4-bit AdamW | 47.4 | 79.0 | 64.1 | 48.0 | 45.2 |
| LLaMA-33B | Original | 54.9 | 79.3 | 58.9 | 45.1 | 42.2 |
| | 32-bit AdamW | 56.4 | 79.2 | 62.6 | 47.1 | 43.8 |
| | 4-bit AdamW | 54.9 | 79.2 | 61.6 | 46.6 | 45.4 |

rather than the optimizer memory consumption alone. In principle, the optimizer states is 2x smaller for 4-bit AdamW than 8-bit AdamW, and about 4x smaller for 4-bit Factor.

The instruction tuning task uses two Nvidia A100 80GB GPUs, while the model is sharded across GPUs with PyTorch's FSDP. In this case, our 4-bit optimizer speeds up training due to reduced communication cost. For the smaller RoBERTa-L and GPT-2 Medium, our 4-bit optimizers appear to be slower than 8-bit AdamW. This is because we have not yet optimize our implementation with fused operators. The speed of our 4-bit optimizers should match or surpass 8-bit optimizers after operator fusion.

We report the largest OPT and LLaMA models trainable under a given memory budget with full-precision and our 4-bit optimizers in Tab. 5. Our optimizer allows for the training of 4x large OPT models, and enables the training of LLaMA-7B model with a single 80GB GPU.

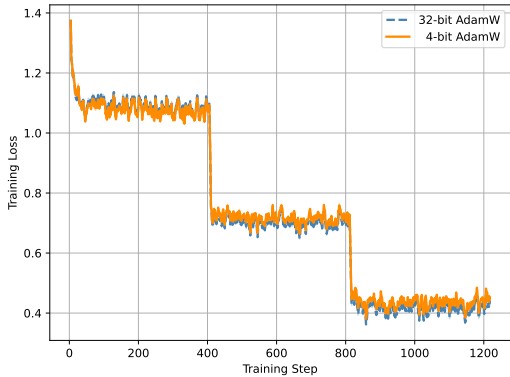

Figure 4: Training loss curve of LLaMA-7B fine-tuning on Alpaca dataset (averaged over 3 runs). Only result of 4-bit AdamW is reported since all parameters are 1-dimension via FSDP packing. See more details in App. D.

**Ablation Study**    Finally, we investigate the effectiveness of our proposed quantization schemes and the sensitivity of each moment to quantization in Tab. 6. We see that quantizing both the first and second moment brings a marginal drop in accuracy. Smaller block size on first moment improves accuracy and takes a step towards lossless performance. Factorizing second moment improves accuracy while leads to better memory efficiency.

## 6   Related Work

**Compression-based memory efficient optimizers**    There have been some works trying to approximate the gradient statistics with sublinear memory cost relative to the number of parameters. Adafactor [46] achieves memory reduction by approximating the second-moment of matrix-shaped

Table 4: Memory and Time of 4-bit optimizers compared with 32-bit AdamW and 8-bit Adam [15].

| Task | Optimizer | Time | Total Mem. | Saved Mem. |
|---|---|---|---|---|
| LLaMA-7B | 32-bit AdamW | 3.35 h | 75.40 GB | 0.00 GB (0%) |
| | 4-bit AdamW | 3.07 h | 31.87 GB | 43.54 GB (57.7%) |
| | 4-bit AdamW (fused) | 3.11 h | 31.88 GB | 43.53 GB (57.7%) |
| RoBERTa-L | 32-bit AdamW | 3.93 min | 5.31 GB | 0.00 GB (0%) |
| | 8-bit AdamW | 3.38 min | 3.34 GB | 1.97 GB (37.1%) |
| | 4-bit AdamW | 5.59 min | 3.02 GB | 2.29 GB (43.1%) |
| | 4-bit AdamW (fused) | 3.17 min | 3.00 GB | 2.31 GB (43.5%) |
| | 4-bit Factor | 4.97 min | 2.83 GB | 2.48 GB (46.7%) |
| GPT-2 Medium | 32-bit AdamW | 2.13 h | 6.89 GB | 0.00 GB (0%) |
| | 8-bit AdamW | 2.04 h | 4.92 GB | 1.97 GB (28.6%) |
| | 4-bit AdamW | 2.43 h | 4.62 GB | 2.37 GB (34.4%) |
| | 4-bit AdamW (fused) | 2.11 h | 4.62 GB | 2.37 GB (34.4%) |
| | 4-bit Factor | 2.30 h | 4.44 GB | 2.45 GB (35.6%) |

Table 5: Largest trainable model under given memory budget. We use a batch size of 1 and max length of 512 for this comparison. FSDP is enabled at GPUs of 80 GB.

| | Largest fine-tunable Model | |
|---|---|---|
| GPU Mem. | 32-bit AdamW | 4-bit AdamW |
| 24 GB | OPT-350M | OPT-1.3B |
| 80 GB | OPT-1.3B | OPT-6.7B |
| 80 GB | - | LLaMA-7B |

Table 6: Ablation study on the impact of compressing different moments to Swin-T pretraining on ImageNet1k.

| Quant. 1st | Quant. 2nd | Factor. 2nd | Acc. |
|---|---|---|---|
| - | - | ✗ | $81.2 \pm 0.05$ |
| B2048/DE | - | ✗ | $80.9 \pm 0.04$ |
| B128/DE | - | ✗ | $81.0 \pm 0.06$ |
| B128/DE | Rank-1/Linear | ✗ | $80.8 \pm 0.02$ |
| B128/DE | Rank-1/Linear | ✓ | $80.9 \pm 0.06$ |

parameters with the outer product of "row" and "column". SM3 [2] considers the cover of parameters and maintain one statistics for each element in the cover. Experimentally, cover composed of slices of co-dimension 1 for each tensor has been adopted. Extreme tensoring [8], compared with SM3, has a different motivation but similar formulation for the experimental choice. These memory efficient methods only focus on memory reduction on second moment and are applicable to most of the second moment based optimizers, like Adagrad [19], Adam [26], AdamW [32], etc.

Another line of work achieves memory reduction by using compression-based method and maintaining coordinate-wise low-bit precision optimizer states. The stability of 16-bit Adam is firstly studied in DALL-E [44]. Dettmers et al. [15] uses block-wise and dynamic exponent quantization to reduce the coordinate-wise optimizer states from 16-bit to 8-bit and is applicable to SGDM and Adam/AdamW. Compression-based methods only reduce memory by a fixed percentage, irrelevant to the number and shape of parameters. Compression-based methods are less memory efficient compared with sublinear memory methods, but exhibit superior performance across a wider range of benchmarks.

**Other memory efficient techniques** Some works focus on the memory efficiency of activations. Activation compressed training [6, 29] keeps low-precision activations in GPU memory via quantization in forward phase and dequantize the low-precision activations to full-precision layer-wisely in backward phase. Gradient checkpointing [7] only keeps activation in a small number of layers in forward phase and recompute the activations in all layers when gradient computation is needed, which leads a trade-off between storage overhead of activations and additional computation cost. These methods can be combined with our optimizer for better memory efficiency. LoRA [24] freezes the pretrained weights and only tunes new initialized low-rank parameters, which greatly reduce the size of computation graph but is only applicable to language model finetuning.

Sharding [40] divides model parameters and optimizer states among multiple devices, allowing for more efficient model training through the use of additional GPUs. Offloading [41, 45] reduces GPU memory consumption by transferring data to CPU memory. Our optimizer can be utilized by these methods to reduce the communication overhead.

# 7 Conclusions, Limitations, and Broader Impact

We propose compression-based memory efficient 4-bit optimizers using quantization and factorization. We observe the heterogeneous outlier patterns in optimizer states and identify the zero-point problem in quantizing second moment as the main bottleneck. 4-bit optimizers achieve lossless performance in finetuning and comparable accuracy in pretraining on a wide range of tasks.

**Limitations**   The optimal quantization setting probably depends on task, datasets, and training details. While rank-1 normalization and linear mapping for second moment quantization performs consistently well in our experiments, task-specific quantization settings not in the scope of the study might perform better and be helpful to achieve lossless performance. Our evaluation is currently limited to language and vision tasks, while the applicability of our method to reinforcement learning, audios, and graph learning tasks still needs further study.

**Broader Impact**   Our work can facilitate the access to large models for pretraining and finetuning, which were previously constrained by GPU memory limitations. This could help democratizing large models and opens up new avenues for research that were previously unattainable due to restricted GPU memory, especially benefiting researchers with limited resources. However, our work can also exaggerate the abuse large models.

## Acknowledgements

This work was supported by the National Key Research and Development Program of China (No. 2021ZD0110502), NSFC Projects (Nos. 62061136001, 62106123, 62076147, U19A2081, 61972224, 62106120), Tsinghua Institute for Guo Qiang, and the High Performance Computing Center, Tsinghua University. J.Z is also supported by the XPlorer Prize.

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

# A Additional Experiment Results

In this section, we show additional experiment results beyond Tab. 2. Tab. 7 shows the results of RoBERTa-L finetuning on each task in GLUE datasets. Tab. 8 shows the results of GPT-2 Medium finetuning on E2E-NLG via different metrics. Tab. 9 shows the EM and F1 of RoBERTa-L finetuning on SQuAD and SQuAD 2.0 datasets.

Table 7: Performance of RoBERTa-Large finetuning on GLUE with diverse optimizers. Medians and std over 5 runs are reported on all tasks. $^{\dagger}$: do not quantize optimizer states for embedding layers; $^{\ddagger}$: $\beta_1 = 0$.

| Optimizer | MNLI | QNLI | QQP | RTE | SST-2 | MRPC | CoLA | STS-B |
|---|---|---|---|---|---|---|---|---|
| 32-bit AdamW | $90.2 \pm 0.00$ | $94.9 \pm 0.00$ | $92.2 \pm 0.00$ | $85.2 \pm 0.14$ | $96.3 \pm 0.00$ | $93.2 \pm 0.01$ | $66.9 \pm 0.01$ | $92.3 \pm 0.00$ |
| 32-bit Adafactor | $90.4 \pm 0.00$ | $94.7 \pm 0.00$ | $92.2 \pm 0.00$ | $85.9 \pm 0.02$ | $96.3 \pm 0.00$ | $92.8 \pm 0.00$ | $67.3 \pm 0.01$ | $92.3 \pm 0.00$ |
| 32-bit Adafactor$^{\ddagger}$ | $90.5 \pm 0.00$ | $94.8 \pm 0.00$ | $92.2 \pm 0.00$ | $87.0 \pm 0.03$ | $96.3 \pm 0.00$ | $92.9 \pm 0.00$ | $68.2 \pm 0.01$ | $92.2 \pm 0.00$ |
| 32-bit SM3 | $90.6 \pm 0.00$ | $94.2 \pm 0.00$ | $89.5 \pm 0.00$ | $85.2 \pm 0.02$ | $96.0 \pm 0.00$ | $90.5 \pm 0.01$ | $62.3 \pm 0.04$ | $91.4 \pm 0.01$ |
| 8-bit AdamW$^{\dagger}$ | $90.4 \pm 0.00$ | $94.8 \pm 0.00$ | $92.2 \pm 0.00$ | $84.8 \pm 0.02$ | $96.2 \pm 0.00$ | $93.2 \pm 0.00$ | $68.0 \pm 0.00$ | $92.2 \pm 0.00$ |
| 4-bit AdamW | $90.2 \pm 0.00$ | $94.5 \pm 0.00$ | $92.0 \pm 0.00$ | $85.2 \pm 0.12$ | $96.3 \pm 0.00$ | $92.8 \pm 0.00$ | $67.3 \pm 0.01$ | $92.5 \pm 0.00$ |
| 4-bit Factor | $90.1 \pm 0.00$ | $94.7 \pm 0.00$ | $92.2 \pm 0.00$ | $85.9 \pm 0.00$ | $96.4 \pm 0.00$ | $92.7 \pm 0.00$ | $68.1 \pm 0.00$ | $92.3 \pm 0.00$ |

Table 8: Performance of GPT-2 Medium finetuning on E2E-NLG Challenge with diverse optimizers. Means and std over 3 runs are reported.

| Optimizer | BLEU | NIST | METEOR | ROUGE-L | CIDEr |
|---|---|---|---|---|---|
| 32-bit AdamW | $67.7 \pm 0.67$ | $8.60 \pm 0.08$ | $45.7 \pm 0.28$ | $68.7 \pm 0.61$ | $2.35 \pm 0.04$ |
| 32-bit Adafactor | $67.2 \pm 0.81$ | $8.61 \pm 0.60$ | $45.3 \pm 0.08$ | $68.3 \pm 0.22$ | $2.35 \pm 0.01$ |
| 32-bit Adafactor$^{\ddagger}$ | $67.2 \pm 0.63$ | $8.54 \pm 0.09$ | $45.6 \pm 0.32$ | $68.5 \pm 0.30$ | $2.32 \pm 0.02$ |
| 32-bit SM3 | $66.9 \pm 0.58$ | $8.59 \pm 0.04$ | $45.4 \pm 0.32$ | $68.2 \pm 0.49$ | $2.33 \pm 0.03$ |
| 8-bit AdamW$^{\dagger}$ | $67.5 \pm 0.87$ | $8.59 \pm 0.08$ | $45.7 \pm 0.52$ | $68.7 \pm 0.97$ | $2.34 \pm 0.06$ |
| 4-bit AdamW | $67.8 \pm 0.51$ | $8.61 \pm 0.08$ | $45.8 \pm 0.23$ | $68.9 \pm 0.33$ | $2.35 \pm 0.07$ |
| 4-bit Factor | $67.6 \pm 0.33$ | $8.59 \pm 0.03$ | $45.6 \pm 0.43$ | $68.6 \pm 0.60$ | $2.34 \pm 0.06$ |

Table 9: Performance of RoBERTa-Large on SQuAD and SQuAD 2.0 with diverse optimizers. Medians and std over 5 runs are reported.

| | SQuAD | | SQuAD 2.0 | |
|---|---|---|---|---|
| Optimizer | EM | F1 | EM | F1 |
| 32-bit AdamW | $89.0 \pm 0.10$ | $94.6 \pm 0.13$ | $85.8 \pm 0.18$ | $88.8 \pm 0.15$ |
| 32-bit Adafactor | $88.8 \pm 0.12$ | $94.6 \pm 0.14$ | $85.8 \pm 0.44$ | $88.7 \pm 0.21$ |
| 32-bit Adafactor$^{\ddagger}$ | $89.0 \pm 0.18$ | $94.7 \pm 0.10$ | $85.9 \pm 0.15$ | $88.8 \pm 0.15$ |
| 32-bit SM3 | $84.2 \pm 0.49$ | $91.7 \pm 0.29$ | $77.2 \pm 0.71$ | $81.1 \pm 0.66$ |
| 8-bit AdamW$^{\dagger}$ | $88.8 \pm 0.15$ | $94.5 \pm 0.04$ | $86.1 \pm 0.26$ | $89.0 \pm 0.26$ |
| 4-bit AdamW | $88.8 \pm 0.08$ | $94.5 \pm 0.10$ | $85.4 \pm 0.28$ | $88.4 \pm 0.26$ |
| 4-bit Factor | $88.8 \pm 0.38$ | $94.6 \pm 0.20$ | $85.9 \pm 0.36$ | $88.9 \pm 0.18$ |

# B Outlier Patterns of Moments

In this section, we give a comprehensive visualization about the outlier pattern of optimizer states. [2] did similar analysis for Adagrad's second moment but here we give a better demonstration about various patterns in optimizer states. The same technique has been applied to parameters and activations in [54].

The outlier pattern of first moment depends on many factors such as data and training hyperparameters. Here we mainly focus on different transformer models and different layers inside. In one transformer block, there is one Attention module and one MLP module, including 6 main parameter matrices. We do not focus on additional parameters including bias and parameters in layer normalization (LN) since they only account a small portion. We denote the 6 matrices by $\mathbf{W}^Q, \mathbf{W}^K, \mathbf{W}^V, \mathbf{W}^O, \mathbf{W}^1, \mathbf{W}^2$, respectively. When the matrices across different layers are involved at the same time, we add a subscript indicating layer index. Note that $\mathbf{W}$ has shape $\mathbb{R}^{C_o \times C_i}$ in a linear layer, we call the output and input dimension by dim0/row and dim1/column, respectively. The per-channel quantizaition used in other works actually correspond to per-row(dim0) normalization here.

**Swin Transformer ImageNet pretraining** In Fig. 5,6,7, the magnitude of first moment in transformer blocks at different depths are shown. It can be seen that the 1-dimensional structure in all parameter matrices are vague at the initial layer. At layer 2, the pattern in $\mathbf{W}^O$ and $\mathbf{W}^1$, that outliers occur at fixed columns, becomes obvious while other parameter matrices remain noisy. At layer 3, the patterns in $\mathbf{W}^O, \mathbf{W}^1, \mathbf{W}^2$ are quite obvious. In $\mathbf{W}^O$ and $\mathbf{W}^2$, the outliers occur at fixed rows while the pattern in $\mathbf{W}^1$ remain unchanged. The 1-dimensional structure in $\mathbf{W}^Q, \mathbf{W}^K, \mathbf{W}^V$ also seem to appear even though not remarkable. It is notable that the pattern of same parameter matrix at different depths are not necessarily same. See $\mathbf{W}^O$ in Fig. 6,7.

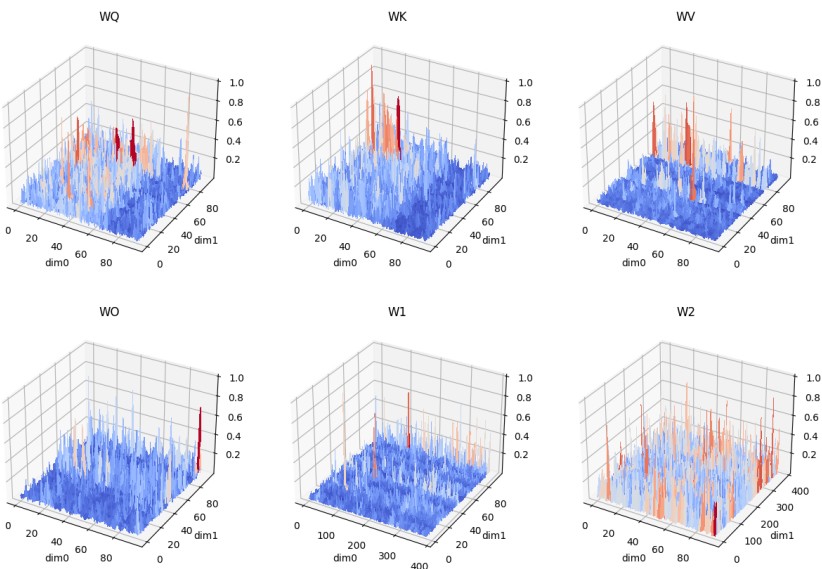

Figure 5: Outlier patterns of first moment in transformer block `layers.0.blocks.0` of Swin-T at epoch 210.

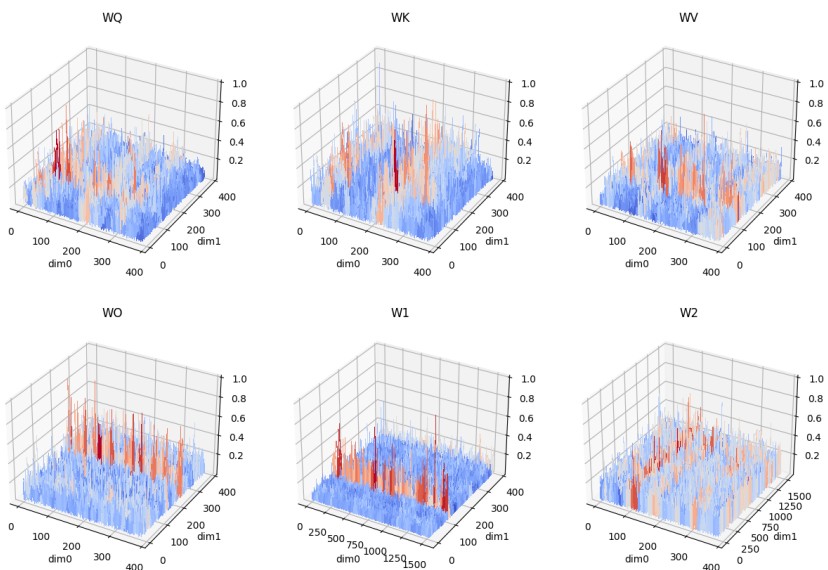

Figure 6: Outlier patterns of first moment in transformer block `layers.2.blocks.0` of Swin-T at epoch 210.

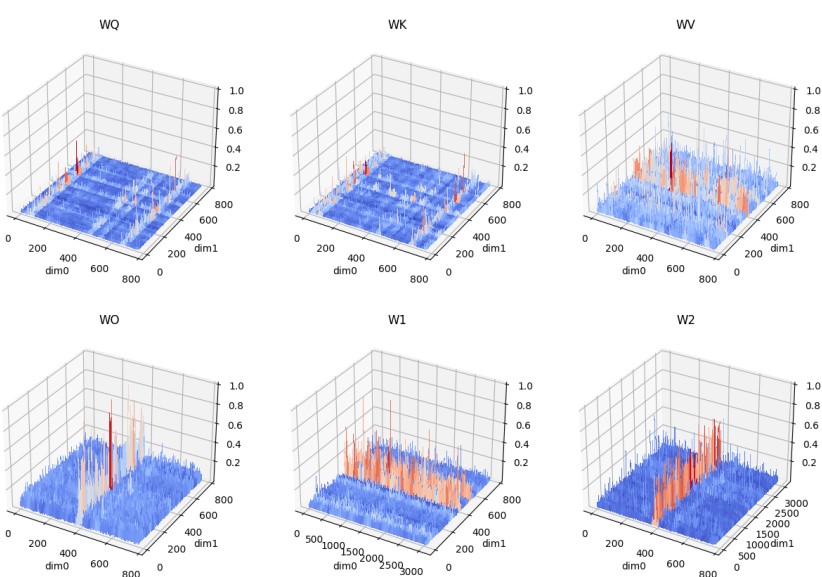

Figure 7: Outlier patterns of first moment in transformer block `layers.3.blocks.0` of Swin-T at epoch 210.

**RoBERTa-Large GLUE finetuning**    In Fig. 8,9,10,11,12,13, the magnitude of first moment in transformer blocks of RoBERTa-Large at different depths are shown. At layer 0 and layer 1 (initial layers), patterns in $\mathbf{W}^O, \mathbf{W}^2$ are obvious. At layer 11 and layer 12 (intermediate layers), patterns are all noisy. At layer 22 and layer 23 (last layers), patterns in $\mathbf{W}^Q, \mathbf{W}^K$ are obvious. Patterns in other matrices are weak.

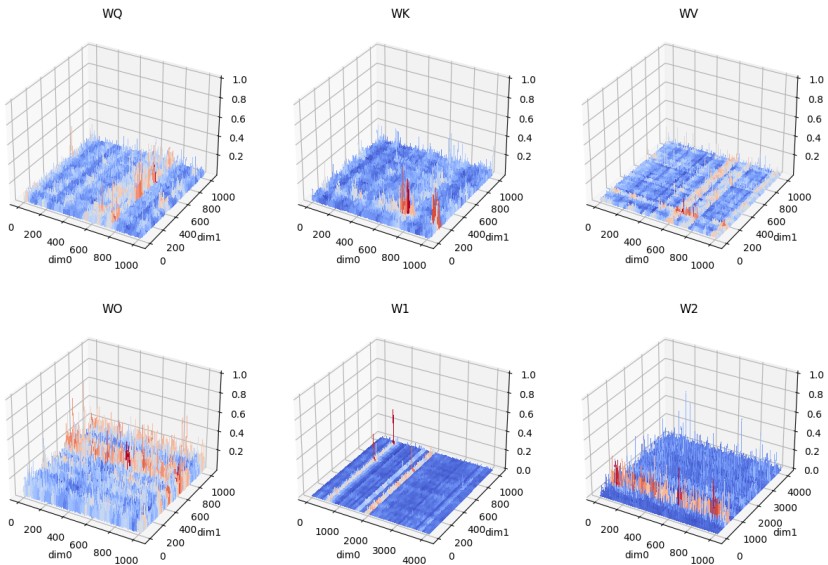

Figure 8: Outlier patterns of first moment in transformer block layer-0 of RoBERTa-Large at epoch 8.

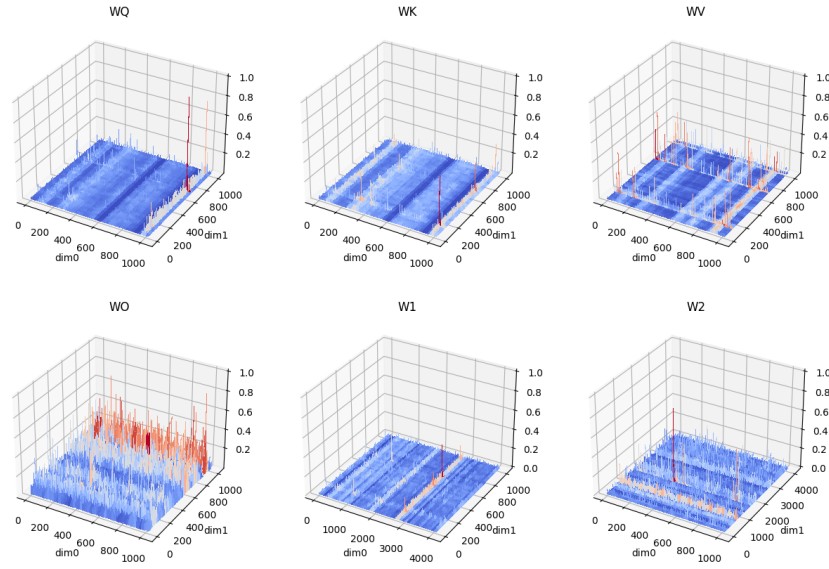

Figure 9: Outlier patterns of first moment in transformer block layer-1 of RoBERTa-Large at epoch 8.

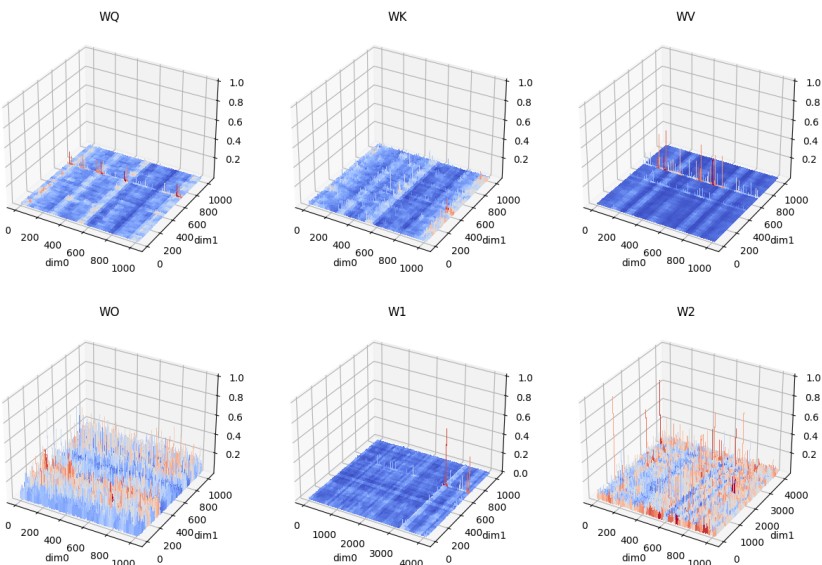

Figure 10: Outlier patterns of first moment in transformer block layer-11 of RoBERTa-Large at epoch 8.

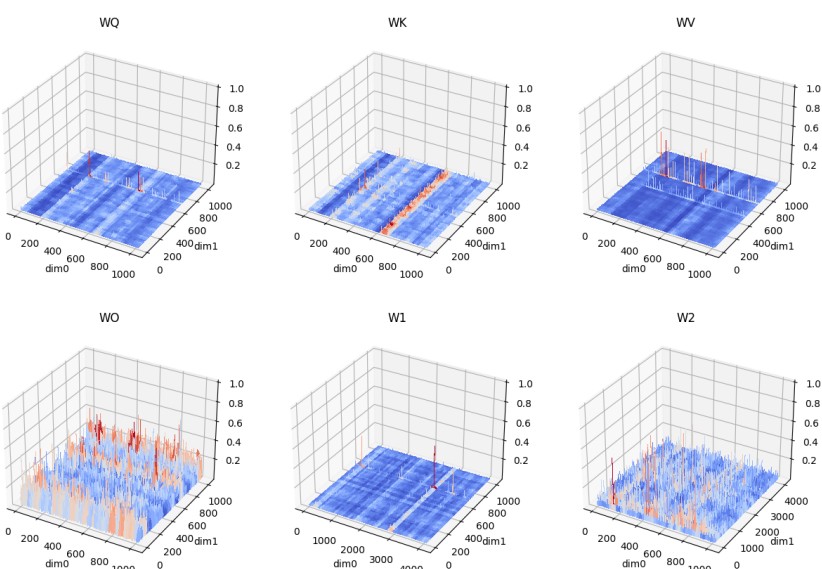

Figure 11: Outlier patterns of first moment in transformer block layer-12 of RoBERTa-Large at epoch 8.

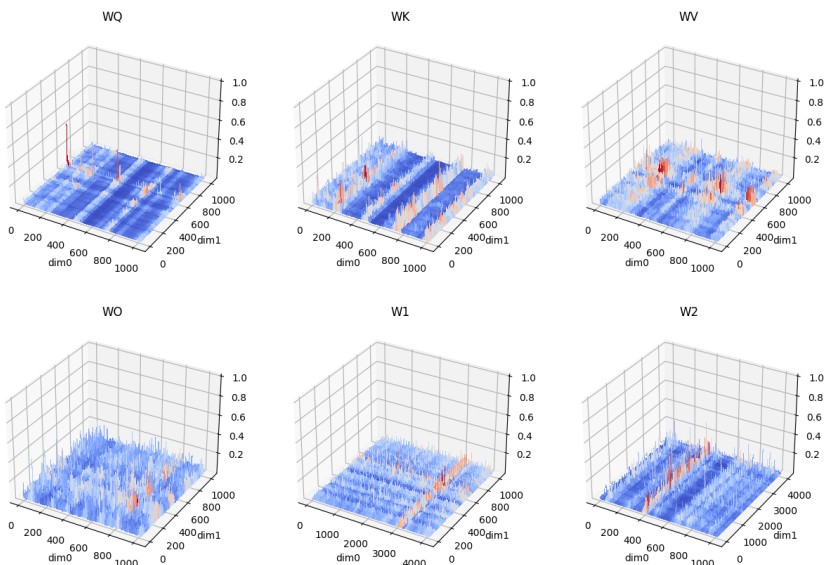

Figure 12: Outlier patterns of first moment in transformer block layer-22 of RoBERTa-Large at epoch 8.

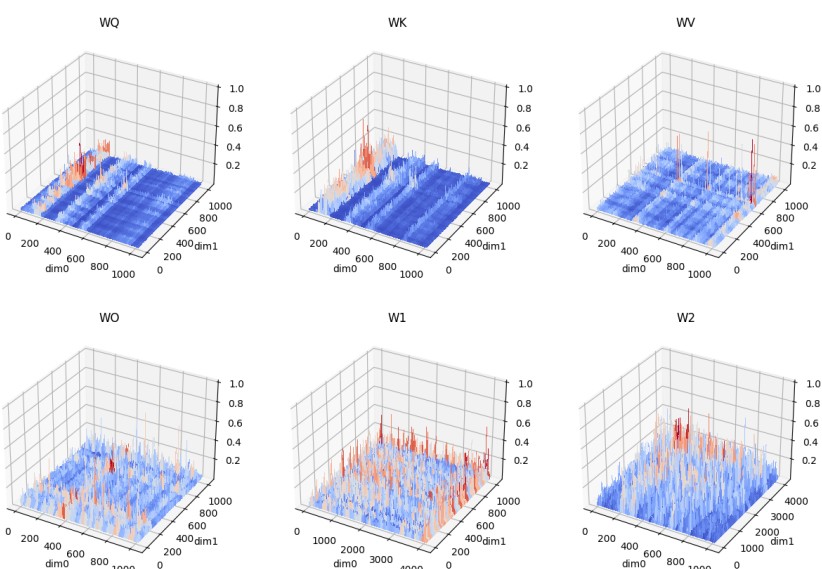

Figure 13: Outlier patterns of first moment in transformer block layer-23 of RoBERTa-Large at epoch 8.

**GPT-2 Medium E2E-NLG finetuning** In Fig. 14,15,16,17,18,19, the magnitude of first moment in transformer blocks of GPT-2 Medium at different depths are shown. At layer 1 and layer 2 (initial layers), patterns in $\mathbf{W}^O$ are obvious. At layer 13 and layer 14 (intermediate layers), patterns in $\mathbf{W}^K, \mathbf{W}^O$ are obvious. At layer 21 and layer 22 (last layers), patterns in $\mathbf{W}^Q, \mathbf{W}^K, \mathbf{W}^V, \mathbf{W}^O$ are obvious. First moment of $\mathbf{W}^1, \mathbf{W}^2$ are consistently noisy throughout layers. It is notable that the rows(or columns) that gather outliers are different across different layers.

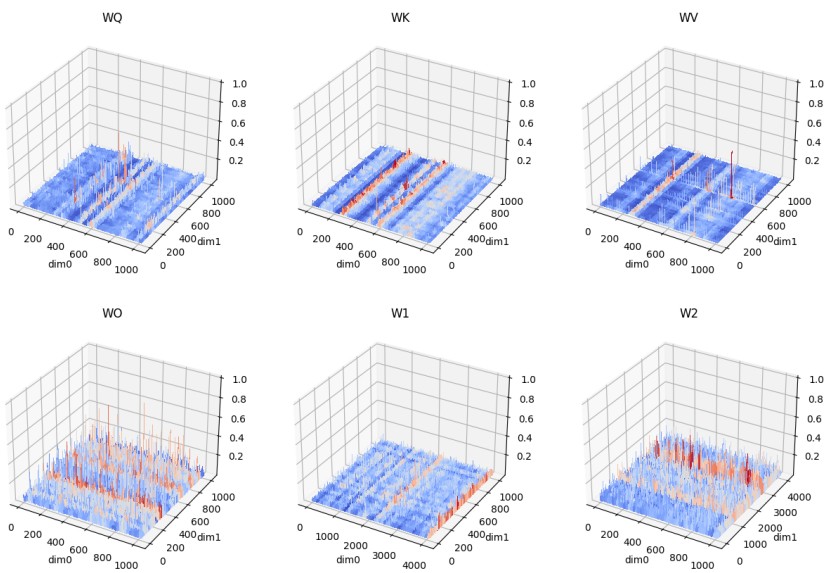

Figure 14: Outlier patterns of first moment in transformer block layer-1 of GPT-2 Medium at epoch 2.

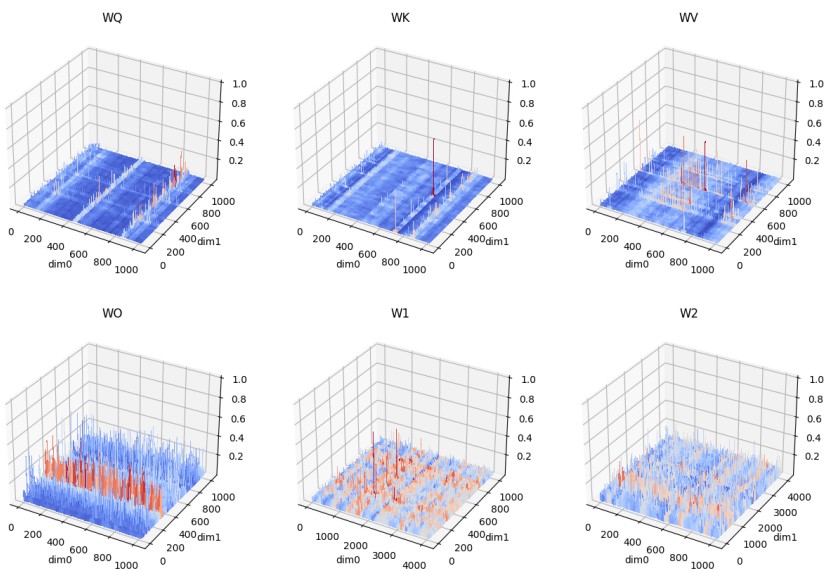

Figure 15: Outlier patterns of first moment in transformer block layer-2 of GPT-2 Medium at epoch 2.

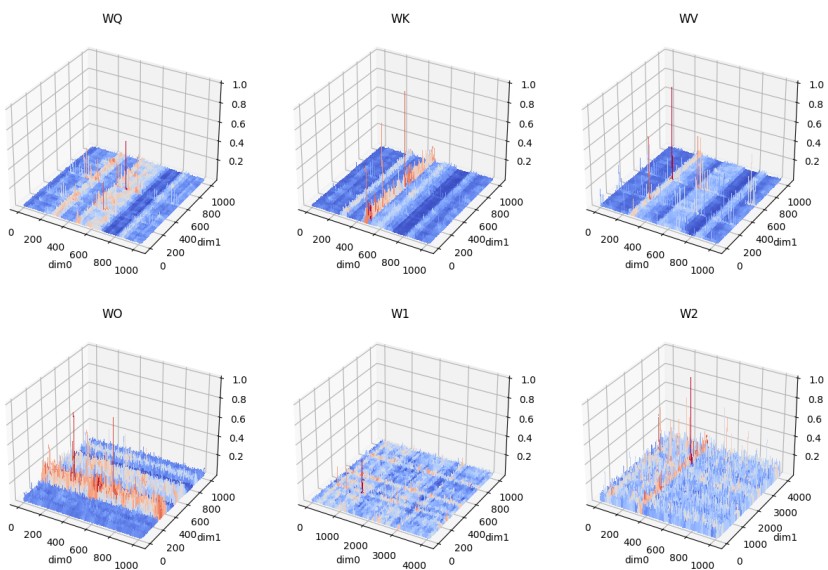

Figure 16: Outlier patterns of first moment in transformer block layer-13 of GPT-2 Medium at epoch 2.

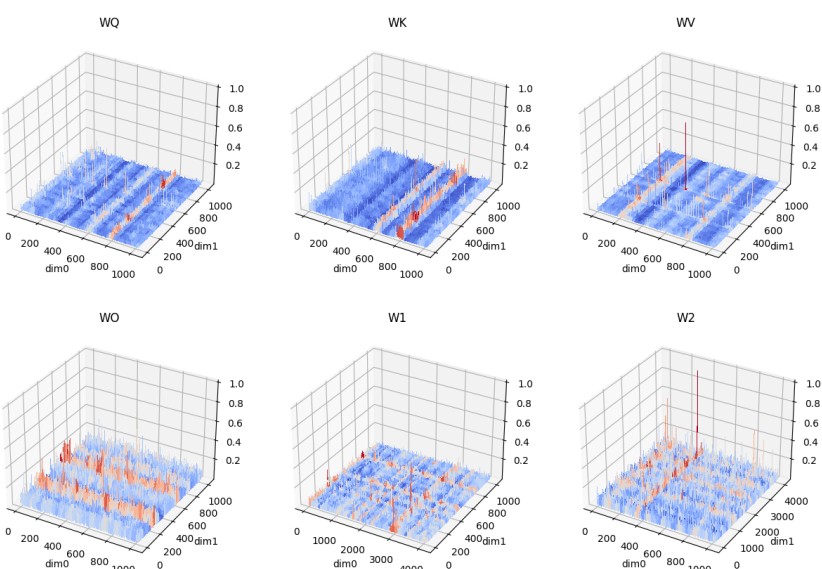

Figure 17: Outlier patterns of first moment in transformer block layer-14 of GPT-2 Medium at epoch 2.

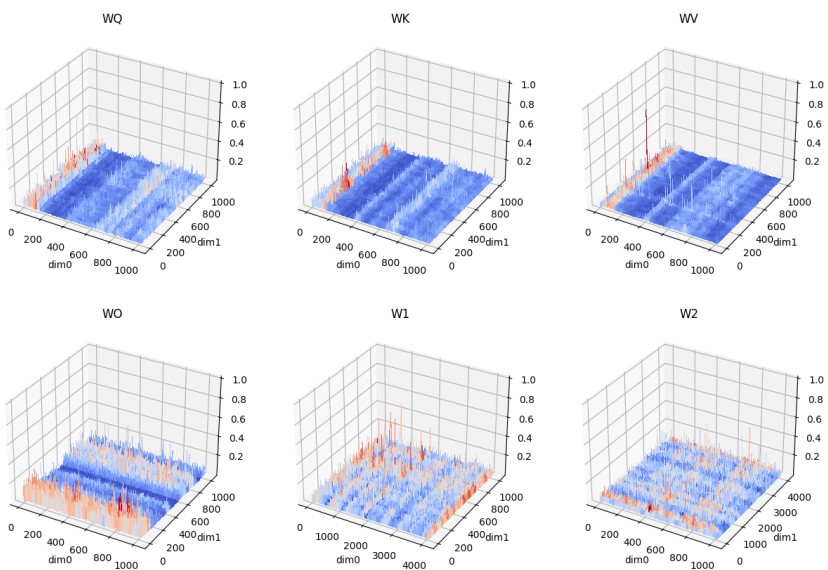

Figure 18: Outlier patterns of first moment in transformer block layer-21 of GPT-2 Medium at epoch 2.

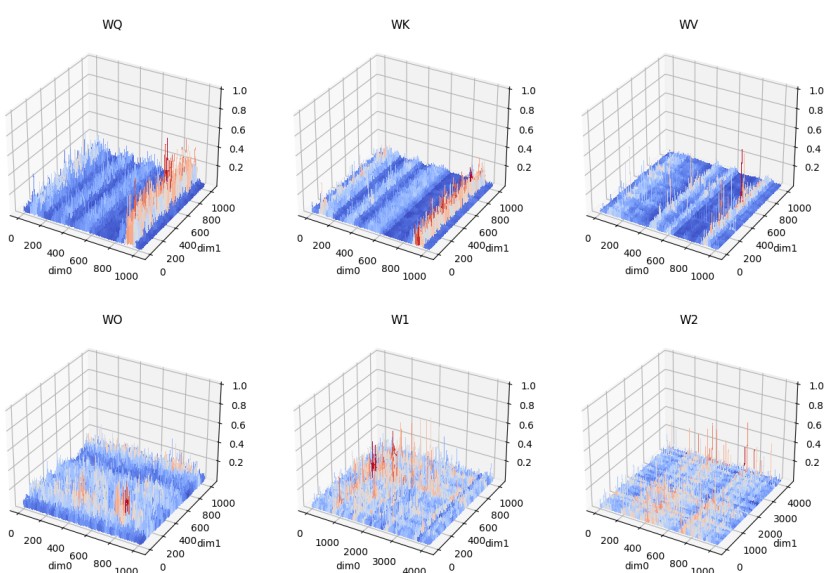

Figure 19: Outlier patterns of first moment in transformer block layer-22 of GPT-2 Medium at epoch 2.

# C  Quantization Quality via Histogram

## C.1  Zero-point Problem

In Fig. 20,21,22, we show the effect of zero-point on quantization error for second moment via histogram. All those figures show the negative impact of zero-point on quantizing second moment. After removing zero-point, the quantization quality improves at a great scale.

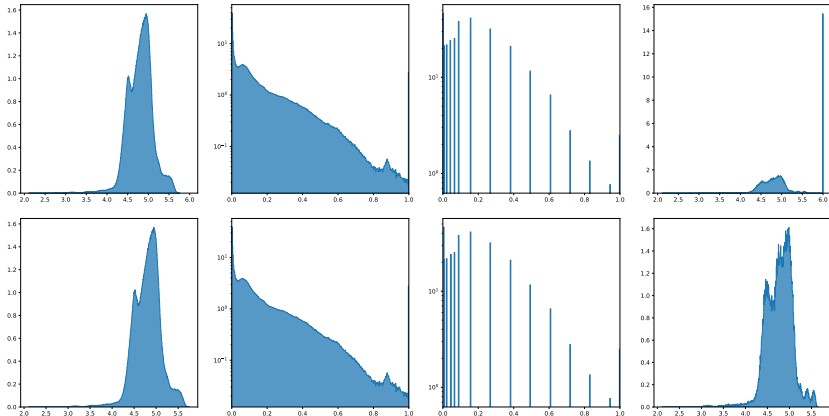

Figure 20: Histogram of second moment of the attention layer ($\mathbf{W}^Q, \mathbf{W}^K, \mathbf{W}^V$) in transformer block-wise layer-2 of GPT-2 Medium at epoch 2. In one horizontal line, the first figure is the original second moment. The second figure is the tensor after normalization. The third figure is the quantized tensor. The last figure is the dequantized object. Both the first and last figure is at log10 scale. Both the second and third take values in [0, 1]. All y-axis represents density. Good quantization methods try to make the third figure identical to the second figure and make the last figure identical to the first figure. Top: B128/DE quantization. Bottom: B128/DE-0 quantization.

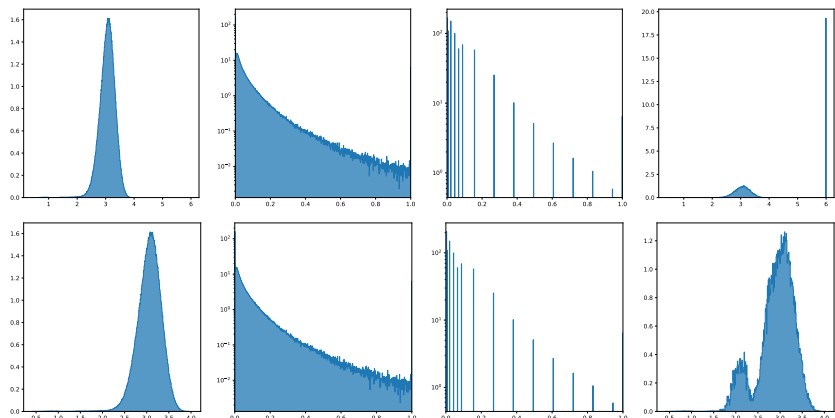

Figure 21: Histogram of second moment of the $\mathbf{W}^V$ in transformer block layer-10 of RoBERTa-Large at epoch 8. Top: B128/DE quantization. Bottom: B128/DE-0 quantization.

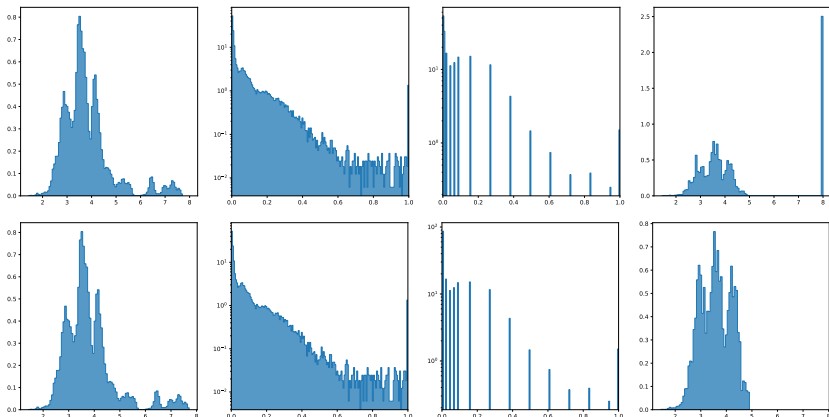

Figure 22: Histogram of second moment of the attention layer ($\mathbf{W}^Q, \mathbf{W}^K, \mathbf{W}^V$) in transformer block `layers.0.blocks.0` of Swin-T at epoch 210. Top: B128/DE quantization. Bottom: B128/DE-0 quantization.

## C.2 Comparison between Block-wise and Rank-1 Normalization

To show the differences in quantization error for second moment between block-wise normalization and rank-1 normalization, some cases where rank-1 normalization approximates better than block-wise normalization are shown in Fig. 23, 25, 27. Also, some cases where rank-1 normalization approximates worse than block-wise normalization is shown in Fig. 24, 26, 28. Empirically, it has been observed that rank-1 normalization yields superior results when the distribution exhibits long-distance multimodal characteristics. On the other hand, block-wise normalization tends to outperform when the distribution displays short-distance multimodal patterns and/or intricate local structures.

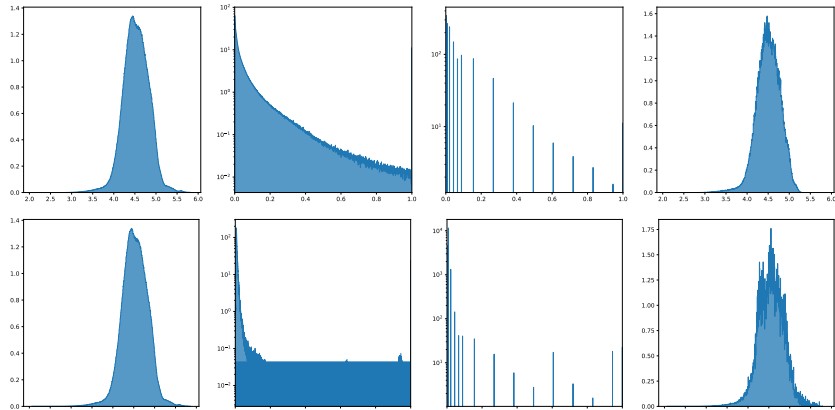

Figure 23: Histogram of second moment of $\mathbf{W}^1$ in transformer block layer-23 of GPT-2 Medium at epoch 2. A case where rank-1 normalization is better than block-wise normalization with block size 128. In this case, the tail in the right side of distribution is captured by rank-1 normalization but lost in block-wise normalization. Top: B128/DE-0 quantization. Bottom: Rank-1/DE-0 quantization.

## C.3 Effectiveness of Block Size in Block-wise Normalization

In Fig. 29,30,31, we show the effect of block size on quantization error for both first and second moments. Fig. 29,30 shows that B2048 normalization quantizes a significant portion of the points to zero, resulting in poor approximation based on the histogram. However, when we utilize a smaller block size of 128, the quantization performance improves. Fig. 31 shows smaller block size improves quantization quality on second moment.

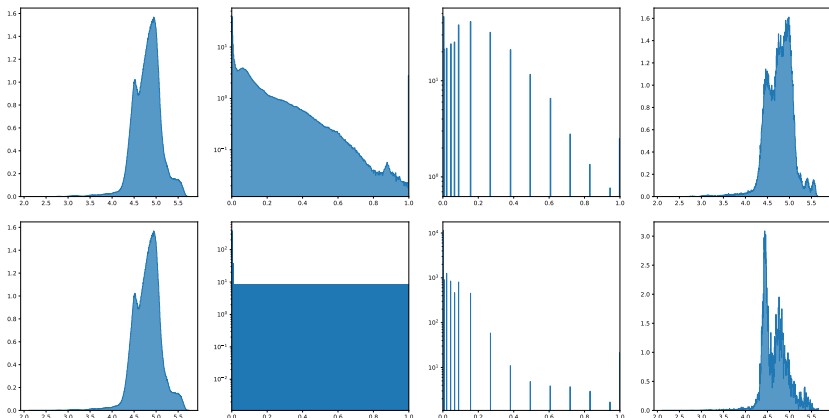

Figure 24: Histogram of second moment of the attention layer ($\mathbf{W}^Q, \mathbf{W}^K, \mathbf{W}^V$) in transformer block layer-2 of GPT-2 Medium at epoch 2. A case where rank-1 normalization is worse than block-wise normalization with block size 128. Top: B128/DE-0 quantization. Bottom: Rank-1/DE-0 quantization.

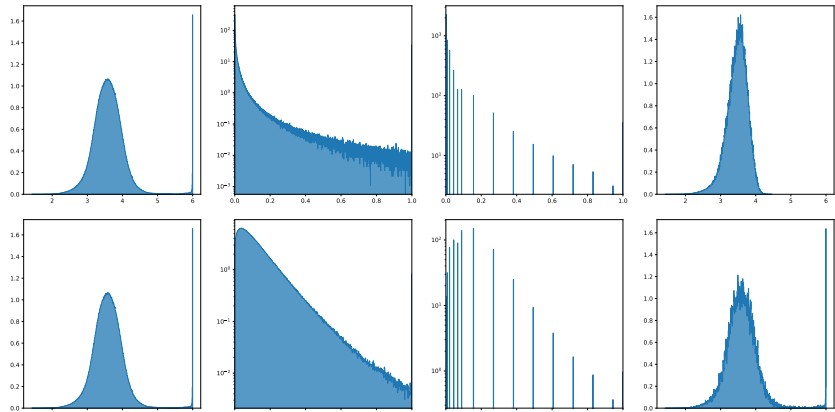

Figure 25: Histogram of second moment of $\mathbf{W}^2$ in transformer block layer-4 of RoBERTa-Large at epoch 8. A case where rank-1 normalization is better than block-wise normalization with block size 128. Top: B128/DE-0 quantization. Bottom: Rank-1/DE-0 quantization.

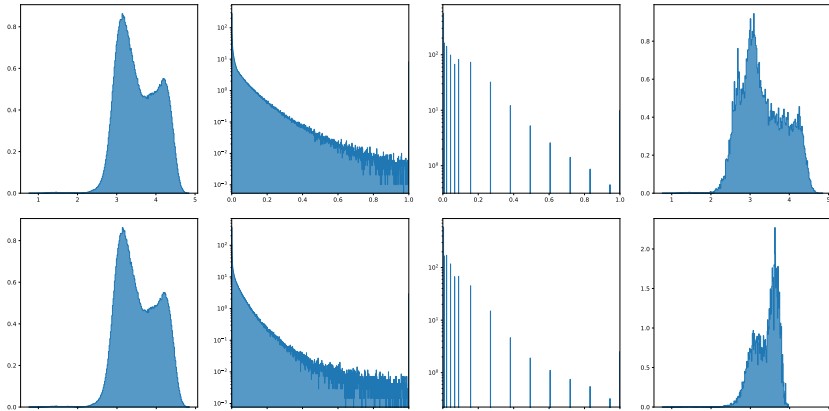

Figure 26: Histogram of second moment of $\mathbf{W}^V$ in transformer block layer-2 of RoBERTa-Large at epoch 8. A case where rank-1 normalization is worse than block-wise normalization with block size 128. Top: B128/DE-0 quantization. Bottom: Rank-1/DE-0 quantization.

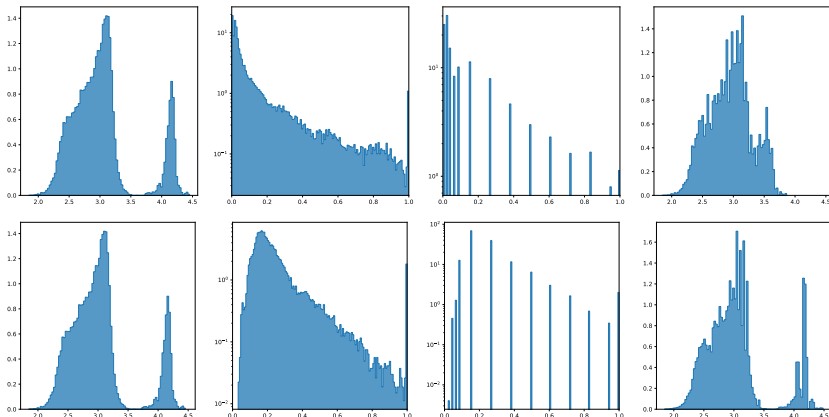

Figure 27: Histogram of second moment of $\mathbf{W}^2$ in transformer block `layers.0.blocks.0` of Swin-T at epoch 210. A case where rank-1 normalization is better than block-wise normalization with block size 128. Top: B128/DE-0 quantization. Bottom: Rank-1/DE-0 quantization.

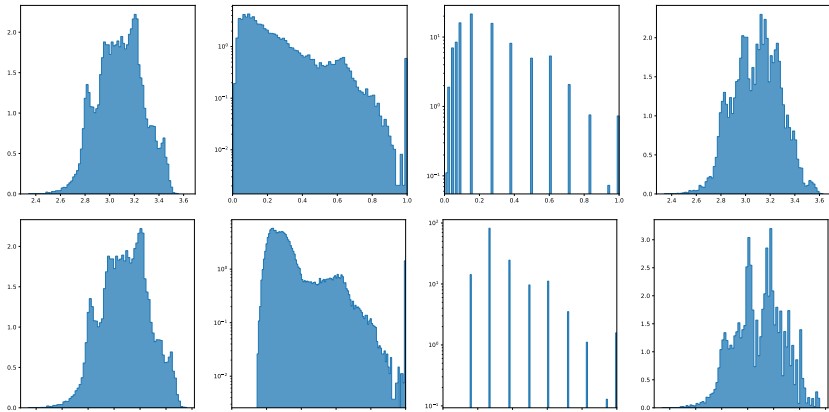

Figure 28: Histogram of second moment of $\mathbf{W}^O$ in transformer block `layers.1.blocks.0` of Swin-T at epoch 210. A case where rank-1 normalization is worse than block-wise normalization with block size 128. Top: B128/DE-0 quantization. Bottom: Rank-1/DE-0 quantization.

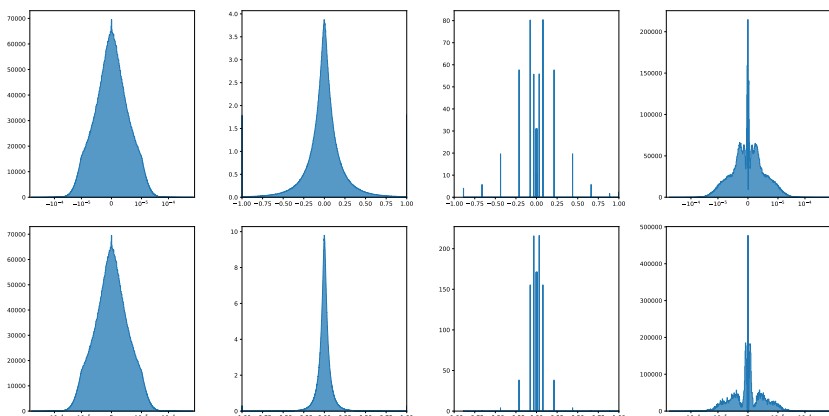

Figure 29: Histogram of first moment of $\mathbf{W}^1$ in transformer block layer-20 of GPT-2 Medium at epoch 2. Top: B128/DE quantization. Bottom: B2048/DE quantization.

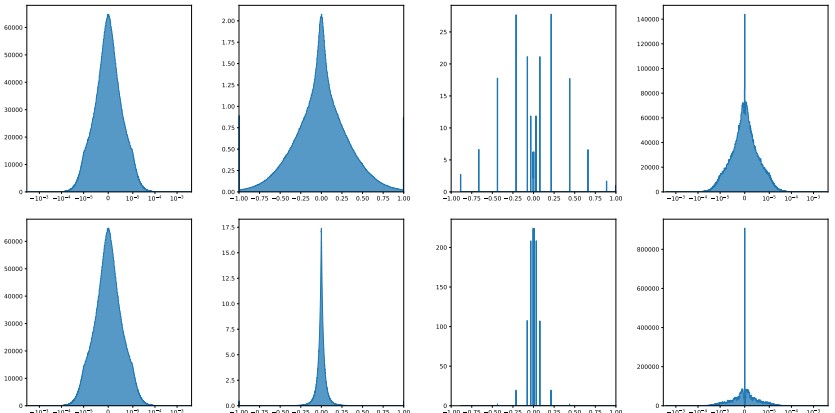

Figure 30: Histogram of first moment of $\mathbf{W}^O$ in transformer block layer-22 of RoBERTa-L at epoch 8. Top: B128/DE quantization. Bottom: B2048/DE quantization.

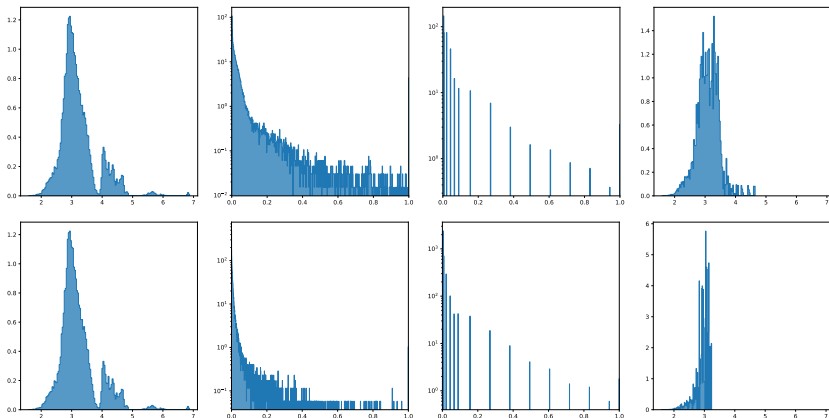

Figure 31: Histogram of second moment of $\mathbf{W}^1$ in transformer block `layers.0.blocks.0` of Swin-T at epoch 210. Top: B128/DE-0 quantization. Bottom: B2048/DE-0 quantization.

# D    Experimental Details

## D.1    Quantization

There are several parameters in deep neural networks that play a delicate role without occupying too much memory, such as normalization layers and bias. In this context, we establish a rule to determine which parameters should not be quantized. For all the experiments we conducted, the rule is straightforward: tensors with a size smaller than or equal to 4096 will not be quantized. However, for larger models with a hidden size exceeding 4096, it is advisable to exclude the bias and normalization layers from quantization. Regarding the quantization settings, as stated in Sec. 5, we employ block-wise normalization with a block size of 128, dynamic exponent mapping for first moment and rank-1 normalization, linear mapping for second moment. When we apply factorization on second moment, only tensors with a dimension greater than or equal to 2 will be factorized while 1-dimensional tensors that meet the specified rule will still be quantized.

8-bit Adam [15] also uses the threshold of 4096 about size to determine whether or not to quantize parameters. Additionally, the implementation in huggingface does not quantize the parameters in `Embedding` layers regardless of the model used. Consequently, we compare our method with the 8-bit Adam that does not quantize `Embedding`.

## D.2 Hyperparameters and Training Details

In each benchmark, unless otherwise specified, we maintain the same hyperparameters for a given optimize across different quantization schemes. Additionally, we use same optimizer hyperparameters across various optimizers, including our 4-bit optimizers, 8-bit Adam [15], SM3 [2], Adafactor [46] and the full precision counterpart AdamW [32]. For Adafactor, we use $\beta_1 > 0$ as default setting which is same as the $\beta_1$ value used in AdamW. Also, the case where $\beta_1 = 0$ is compared. The other newly introduced hyperparameters in Adafactor are set to their default values and remain fixed throughout the experiments. For SM3, we compare with the $\beta_1 > 0$ configuration, same as the $\beta_1$ value used in AdamW.

Table 10: The hyperparameters for RoBERTa-L fine-tuning on GLUE.

| Dataset | MNLI | QNLI | QQP | RTE | MRPC | SST-2 | CoLA | STS-B |
|---|---|---|---|---|---|---|---|---|
| Batch Size | 32 | 32 | 32 | 16 | 16 | 32 | 16 | 16 |
| LR | 1e-5 | 1e-5 | 1e-5 | 2e-5 | 1e-5 | 1e-5 | 1e-5 | 2e-5 |
| Warmup | 7432 | 1986 | 28318 | 122 | 137 | 1256 | 320 | 214 |
| Max Train Steps | 123873 | 33112 | 113272 | 2036 | 2296 | 20935 | 5336 | 3598 |
| Max Seq. Len. | 128 | 128 | 128 | 512 | 512 | 512 | 512 | 512 |

Table 11: The hyperparameters for RoBERTa-L fine-tuning on SQuAD and SQuAD 2.0.

| Dataset | SQuAD & SQuAD 2.0 |
|---|---|
| Batch Size | 48 |
| LR | 1.5e-5 |
| # Epochs | 2 |
| Warmup Ratio | 0.06 |
| Max Seq. Len. | 384 |

Table 12: The hyperparameters for GPT-2 on E2E.

| Dataset | E2E |
|---|---|
| | Training |
| Batch Size | 8 |
| LR | 4e-5 |
| # Epochs | 5 |
| Warmup | 500 |
| Max Seq. Len. | 512 |
| Label Smooth | 0.1 |
| | Inference |
| Beam Size | 10 |
| Length Penalty | 0.8 |
| no repeat ngram size | 4 |

**RoBERTa** We train all of our RoBERTa-L models with PyTorch Huggingface[††]. On GLUE benchmark, we mainly follow the hyperparameters in fairseq [36]. We use $\beta_1 = 0.9$, $\beta_2 = 0.98$, $\epsilon = $ 1e-6, a weight decay factor of 0.1 and linear learning rate schedule. Other hyperparameters are listed in Tab. 10. On SQuAD benchmark, we mainly follow the reported hyperparameters in RoBERTa paper [30]. We use $\beta_1 = 0.9$, $\beta_2 = 0.98$, $\epsilon = $ 1e-6, a weight decay factor of 0.01 and linear learning rate schedule. The other hyperparameters are listed in Tab. 11. On both datasets, we report the median and standard deviation results over 5 runs, the result in each run is taken from the best epoch. We utilize single RTX 3090 or 4090 GPU for runs of each task in GLUE datasets and four RTX 3090 or 4090 GPUs for SQuAD and SQuAD 2.0.

---

[††]https://github.com/huggingface/transformers

On SQuAD 2.0, there may be a performance gap observed between the reproduced results using 32-bit AdamW and the original results reported in the original paper. This is because there are some questions without answers in SQuAD 2.0. It is worth noting that the approach employed by Liu et al. [30] to handle unanswered questions may differ from the solutions utilized in the BERT paper [17], which is the reference implementation we are using

**GPT-2**   We train all of our GPT-2 Medium models with the LoRA codebase[‡‡]. We mainly follow the hyperparameters in [28] and [24]. We use $\beta_1 = 0.9$, $\beta_2 = 0.999$, $\epsilon = 1\text{e-}6$, a weight decay factor of 0.01 and linear learning rate schedule. The other hyperparameters used in GPT-2 are listed in Tab. 12. We report the mean and standard deviation results over 3 runs, the result in each run is taken from the best epoch. We utilize fours RTX 3090 or 4090 GPUs for runs of this task.

**Transformer**   We train all of our Transformer-Base models for machine translation with codebase[§§]. We completely follow the hyperparameters in the codebase. We report the mean and standard deviation results over 3 runs, the result in each run is taken from the best epoch. We utilize eight RTX 3090 or 4090 GPUs for runs of this task.

**Swin**   We train all of our Swin-T models with its official codebase[¶¶]. We completely follow the hyperparameters in the codebase. We report the mean and standard deviation results over 3 runs, the result in each run is taken from the best epoch. We utilize eight RTX 3090 or 4090 GPUs for runs of this task.

**LLaMA**   We fine-tune LLaMA-7B, LLaMA-13B and LLaMA-33B with Alpaca codebase[***]. We follow the hyperparameters in the codebase for LLaMA-7B and LLaMA-13B, and the hyperparameters of LLaMA-33B are consistent with LLaMA-13B. We fine-tune LLaMA-7B with two A100 80GB GPUs. The training loss curve is the mean results over 3 runs. For LLaMA-7B, we enable Fully Sharded Data Parallelism (FSDP), which packs parameters into 1-dimensional array. This packing process makes it difficult to apply factorization directly without additional engineering efforts. Consequently, we only compare the performance of 4-bit AdamW with its full precision counterpart.

### D.3   Memory and Computing Efficiency

In Tab. 4, we present measurements of memory usage in practical settings, i.e. training configuration described in Sec. D.2. Specifically, we measure the memory usage for LLaMA-7B using 2 A100 80G GPUs, RoBERTa-L using 1 RTX 4090 GPU, and GPT-2 Medium using 4 RTX 4090 GPUs. Additionally, the time measurement for RoBERTa-L is conducted on the RTE task.

## E   Quantization Formulation Details

### E.1   Signed Case

In this section, we discuss quantization function for signed tensors and the differences compared to unsigned case. Regarding the normalization operator, the only difference lies in the fact that the sign of the tensor remains unchanged before and after normalization. Formally, let $\mathbf{N}$ be the normalization operator for the unsigned cases. For the signed case, the normalization can be defined as

$$n_j := \text{sign}(x_j)\mathbf{N}(|x_j|).$$

Therefore, the unit interval for signed case is [-1, 1]. Regarding the mapping operator, the difference lies in the values of quantization mappings. See App. E.2 for more details.

### E.2   Quantization Mappings

In this work, we mainly consider linear mapping and dynamic exponent mapping [13]. See Fig. 32 for illustration of quantization mappings.

---

[‡‡]https://github.com/microsoft/LoRA

[§§]https://github.com/NVIDIA/DeepLearningExamples/tree/master/PyTorch/Translation/Transformer

[¶¶]https://github.com/microsoft/Swin-Transformer

[***]https://github.com/tatsu-lab/stanford_alpaca

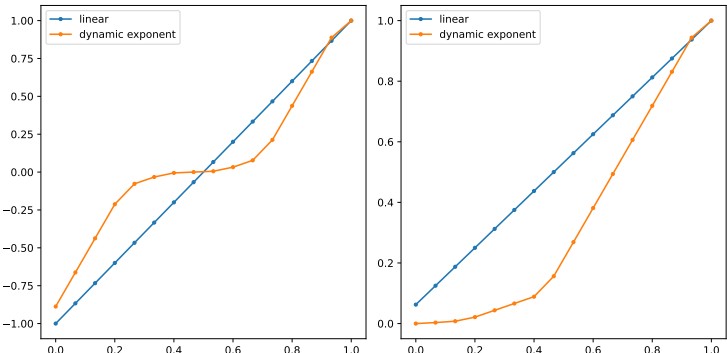

Figure 32: Visualization of the quantization mappings for the linear and dynamic exponent at 4-bit precision. Left: Signed case. Right: Unsigned case.

**Linear mapping**  It is notable that the linear mapping considered in our work does not include zero in both signed case and unsigned case. Actually, we only use linear mapping in unsigned case, which is defined as `torch.linspace(0, 1, (2 ** b) + 1)[1:]`.

**Dynamic exponent mapping**  Let $b$ be the total bits. In the main text, we mentioned that dynamic exponent takes the form $\mathbf{T}(i) = 10^{-E(i)}\text{fraction}(i)$. In following paragraphs, we will define the dynamic exponent mapping formally based on the binary representation.

In unsigned case, dynamic exponent mapping [13] is composed of exponent bits $E$, one indicator bit and fraction bits $F$, where $b = 1 + E + F$. It uses the number of leading zero bits $E$ represents the exponent with base 10. The first bit, which is one, serves as an indicator bit that separates the exponent and the unsigned linear fraction. The remaining bits $F$ represent an unsigned linear fraction distributed evenly in $(0.1, 1)$, which is formally defined as

$$p_j = \frac{1 - 0.1}{2^F}j + 0.1, \quad 0 \le j \le 2^F,$$

$$\text{fraction}[k] = \frac{p_k + p_{k+1}}{2}, \quad 0 \le k \le 2^F - 1.$$

Therefore, a number with $E$ exponent bits and $F$ fraction bits valued $k$ has a value of

$$10^{-E} \times \text{fraction}[k].$$

For signed case, the only difference is that dynamic exponent mapping additionally uses the first bit as the sign thus we have $b = 1 + E + 1 + F$. Specially, at 8-bit case, we learn from the codebase[†††] that dynamic exponent mapping assign $00000000_2 = 0_{10}$, $00000001_2 = 1_{10}$ in unsigned case and assign $10000000_2$ and $00000000_2$ with $1_{10}$ and $0_{10}$, respectively. This means $-1_{10}$ is not defined and the mapping is not symmetric in signed case. Finally, after collecting all the represented numbers and arranging them in a sorted, increasing list, which has a length of $2^b$, the quantization mapping $\mathbf{T}(i)$ returns the $i$-th element of this list.

The construction of dynamic exponent mapping is unrelated to the number of bits. Therefore, when we say we barely turn the 8-bit optimizer into 4-bit optimizer, it just use 4 total bits. The corner cases mentioned in last paragraph remain unchanged.

### E.3  Stochastic Rounding

Stochastic rounding is only used in Tab. 1. In this section, we talk about how to integrate stochastic rounding into our formulation of quantization. When stochastic rounding is used, the definition of mapping $\mathbf{M}$ has some minor changes. Specifically, $\mathbf{M}$ is still an element-wise function and defined as

$$\mathbf{M}(n_j) = \arg \min_{0 \le i < 2^b} \{n_j - \mathbf{T}(i) : n_j - \mathbf{T}(i) \ge 0\} \cup \arg \max_{0 \le i < 2^b} \{n_j - \mathbf{T}(i) : n_j - \mathbf{T}(i) \le 0\}.$$

---

[†††] https://github.com/TimDettmers/bitsandbytes

In other words, $\mathbf{M}$ maps each entry $n_j$ to the maximal index set $\mathbf{M}(n_j)$ such that for any $i \in \mathbf{M}(n_j)$ there is no other $0 \le k \le 2^b - 1$ with $\mathbf{T}(k)$ lying between $\mathbf{T}(i)$ and $n_j$. Actually, $\mathbf{M}$ acts as a filter of $\mathbf{T}$ and give a more fine-grained range of quantized output candidates. In this definition, $\mathbf{M}(n_j)$ has only one or two points since only stochastic rounding is considered in the final step.

Finally, we define stochastic rounding $\mathbf{R}_s$. When $\mathbf{M}(n_j)$ only has one point, $\mathbf{R}_s$ just output this point. When $\mathbf{M}(n_j)$ has two points $q_1$ and $q_2$ with $\mathbf{T}(q_1) < n_j < \mathbf{T}(q_2)$, stochastic rounding ($\mathbf{R}_s$) is defined as

$$
\mathbf{R}_s\left(n_j, q_1, q_2\right) = 
\begin{cases}
q_2, \text{with proba. } \frac{n_j - \mathbf{T}(q_1)}{\mathbf{T}(q_2) - \mathbf{T}(q_1)} \\[3mm]
q_1, \text{with proba. } \frac{\mathbf{T}(q_2) - n_j}{\mathbf{T}(q_2) - \mathbf{T}(q_1)}
\end{cases}
$$

## F   Compression-based Memory Efficient Optimizer Instances

In this section, we present some examples about compression-based memory efficient optimizers. See Compression-based Memory Efficient SGDM in Alg. 2 and Adam in Alg. 3.

---

**Algorithm 2** Compression-based Memory Efficient SGDM

---

**Require:** initial parameter $\theta_0 \in \mathbb{R}^p$, learning rate $\alpha$, initial first moment $\bar{m}_0 = 0$, total number of iterations $T$ and momentum parameter $\beta$.

1: **for** $t = 1, 2, \ldots, T$ **do**
2:     Sample a minibatch $\zeta_t$ and get stochastic gradient $g_t = \nabla_\theta f(\theta_{t-1}, \zeta_t)$
3:     $m_{t-1} \leftarrow \text{decompress}(\bar{m}_{t-1})$
4:     $m_t \leftarrow \beta \cdot m_{t-1} + g_t$
5:     $\theta_t \leftarrow \theta_{t-1} - \alpha \cdot m_t$
6:     $\bar{m}_t \leftarrow \text{compress}(m_t)$
7: **end for**
8: **return** $\theta_T$

---

---

**Algorithm 3** Compression-based Memory Efficient Adam

---

**Require:** initial parameter $\theta_0 \in \mathbb{R}^p$, learning rate $\alpha$, initial moments $\bar{m}_0 = 0, \bar{v}_0 = 0$, total number of iterations $T$ and hyperparameters $\beta_1, \beta_2, \epsilon$.

1: **for** $t = 1, 2, \ldots, T$ **do**
2:     Sample a minibatch $\zeta_t$ and get stochastic gradient $g_t = \nabla_\theta f(\theta_{t-1}, \zeta_t)$
3:     $m_{t-1}, v_{t-1} \leftarrow \text{decompress}(\bar{m}_{t-1}), \text{decompress}(\bar{v}_{t-1})$
4:     $m_t \leftarrow \beta_1 \cdot m_{t-1} + (1 - \beta_1) \cdot g_t$
5:     $v_t \leftarrow \beta_2 \cdot v_{t-1} + (1 - \beta_1) \cdot g_t^2$
6:     $\hat{m}_t \leftarrow m_t / (1 - \beta_1^t)$
7:     $\hat{v}_t \leftarrow v_t / (1 - \beta_2^t)$
8:     $\theta_t \leftarrow \theta_{t-1} - \alpha \cdot \hat{m}_t / (\sqrt{\hat{v}_t} + \epsilon)$
9:     $\bar{m}_t, \bar{v}_t \leftarrow \text{compress}(m_t), \text{compress}(v_t)$
10: **end for**
11: **return** $\theta_T$

---

## G   Rank-1 Normalization

In this section, we present the detailed formulation of rank-1 normalization in Alg. 4.

## H   Theoretical Analysis

The convergence of low-bit optimizers can be guaranteed if their fp32 counterparts converge. Here we provide a theorem about the convergence of quantized SGDM (Alg. 2) under some assumptions. We believe the convergence of low-bit AdamW could be inferred from the convergence of AdamW.

---
**Algorithm 4** Rank-1 Normalization

---
**Require:** tensor $x \in \mathbb{R}^{d_1 \times \cdots \times d_p}$; statistics $\mu_r \in \mathbb{R}^{d_r}$ for $1 \leq r \leq p$; permutation function $\Phi$ mapping $\{1, \ldots, d\}$ to indices of tensor $x$, where $d = d_1 \times \cdots \times d_p$.
1: **for** $r = 1, 2, \ldots, p$ **do**
2:    **for** $j = 1, 2, \ldots, d_r$ **do**
3:       $\mu_{r,j} = \max_{i_1,\ldots,i_{r-1},i_{r+1},\ldots,i_p} \left| x_{[i_1,\ldots,i_{r-1},j,i_{r+1},\ldots,i_p]} \right|$
4:    **end for**
5: **end for**
6: **for** $i = 1, 2, \ldots, d$ **do**
7:    $M_i = \min_{1 \leq r \leq p} \mu_{r,\Phi(i)_r}$
8: **end for**
9: reshape 1-dimensional array $M$ to the same shape as $x$
10: **return** $x/M$

---

First, we make some assumptions. The first three are rather standard in stochastic optimization literature, while last two depict properties of stochastic quantizers.

1. *(Convexity) The objective function is convex and has an unique global minimum $f(\theta^*)$.*
2. *(Smoothness) The objective $f(\theta)$ is continuous differentiable and L-smooth;*
3. *(Moments of stochastic gradient) The stochastic gradient $g$ is unbiased, i.e., $\mathbb{E}[g(\theta)] = \nabla f(\theta)$, and has bounded variance, i.e., $\mathbb{E}\left[ \|g(\theta) - \nabla f(\theta)\|^2 \right] < \sigma^2$, $\forall \theta \in \mathbb{R}^d$.*
4. *(Unbiased quantizer) $\forall x \in \mathbb{R}^d$, $\mathbb{E}[Q(x)] = x$.*
5. *(Bounded quantization variance) $\forall x \in \mathbb{R}^d$, $\mathbb{E}\left[ \|Q_m(x) - x\|^2 \right] \leq \sigma_m^2$.*

Then, we have following theorem:

**Theorem 1.** *Consider the Algorithm 2 with Assumptions 1-5. Let $\alpha \in (0, \frac{1-\beta}{L}]$, then for all $T > 0$ we have*

$$\mathbb{E}[f(\bar{\theta}_T) - f_*] \leq \frac{1}{2T} \left( \frac{L\beta}{1-\beta} + \frac{1-\beta}{\alpha} \right) \|\theta_0 - \theta_*\|^2$$
$$+ \frac{\alpha\sigma^2}{(1-\beta)} + \frac{\alpha\sigma_m^2}{(1-\beta)}. \tag{2}$$

*where $\bar{\theta}_T = \frac{1}{T} \sum_{i=0}^{T-1} \theta_i$.*

### H.1 Proof of Theorem 1

To prove Theorem 1, we need some useful lemmas.

**Lemma 1.** *In Algorithm 2, The conditional first and second moments of $g_t$ satisfies*

$$\mathbb{E}[g_t|\theta_{t-1}] = \nabla f(\theta_{t-1}) \tag{3}$$
$$\mathbb{E}\left[ \|g_t\|^2 |\theta_{t-1} \right] \leq \|\nabla f(\theta_{t-1})\|^2 + \sigma^2 \tag{4}$$

*Proof.* By assumption, we easily have

$$\mathbb{E}\left[ g_t|\theta_{t-1} \right] = \nabla f(\theta_{t-1}).$$

With Assumption 3, it holds true that

$$\mathbb{E}\left[ \|g(\theta)\|^2 \right] = \mathbb{E}\left[ \|g(\theta) - \nabla f(\theta) + \nabla f(\theta)\|^2 \right]$$
$$= \mathbb{E}\left[ \|g(\theta) - \nabla f(\theta)\|^2 \right] + \mathbb{E}\left[ \|\nabla f(\theta)\|^2 \right] + 2\mathbb{E}\left[ \langle g(\theta) - \nabla f(\theta), \nabla f(\theta) \rangle \right]$$
$$= \mathbb{E}\left[ \|g(\theta) - \nabla f(\theta)\|^2 \right] + \mathbb{E}\left[ \|\nabla f(\theta)\|^2 \right]$$
$$\leq \sigma^2 + \|\nabla f(\theta)\|^2,$$

which implies the second part. $\qquad \square$

**Lemma 2.** *If Assumptions 3-5 hold, then sequence $\{z_t\}$ satisfies*

$$z_{t+1} - z_t = \frac{1}{1-\beta}(\theta_{t+1} - \theta_t) - \frac{\beta}{1-\beta}(\theta_t - \theta_{t-1}) \tag{5}$$

$$\mathbb{E}[z_{t+1} - z_t] = \frac{-\alpha}{1-\beta}\nabla f(\theta_t) \tag{6}$$

$$\mathbb{E}[\|z_{t+1} - z_t\|^2] \leq 2\left(\frac{\alpha}{1-\beta}\right)^2\left(\mathbb{E}[\|g_{t+1}\|^2] + \sigma_m^2\right). \tag{7}$$

*Proof.* By definition of $z_t$, we have the first equation immediately. Take expectation on the first equation and we get

$$\mathbb{E}[z_{t+1} - z_t] = \frac{1}{1-\beta}\mathbb{E}[\theta_{t+1} - \theta_t] - \frac{\beta}{1-\beta}\mathbb{E}[\theta_t - \theta_{t-1}].$$

Note that

$$\begin{aligned}
\mathbb{E}[\theta_{t+1} - \theta_t] &= \mathbb{E}[\theta_{t+1} - (\theta_t - \alpha m_{t+1})] - \mathbb{E}[\alpha m_{t+1}] \\
&= -\alpha\mathbb{E}[m_{t+1}] \\
&= -\alpha\mathbb{E}[\beta m_t + g_{t+1}] \\
&= -\alpha\beta\mathbb{E}[m_t] - \alpha\nabla f(\theta_t),
\end{aligned}$$

and

$$\begin{aligned}
\mathbb{E}[\theta_t - \theta_{t-1}] &= \mathbb{E}[\theta_t - (\theta_{t-1} - \alpha m_t)] - \mathbb{E}[\alpha m_t] \\
&= -\alpha\mathbb{E}[m_t],
\end{aligned}$$

which gives the second equation.

$$\mathbb{E}[z_{t+1} - z_t] = \frac{-\alpha}{1-\beta}\nabla f(\theta_t)$$

For the last equation, since

$$\begin{aligned}
z_{t+1} - z_t &= \frac{1}{1-\beta}(\theta_{t+1} - \theta_t) - \frac{\beta}{1-\beta}(\theta_t - \theta_{t-1}) \\
&= -\frac{\alpha}{1-\beta}(m_{t+1} - \beta m_t)
\end{aligned}$$

Take expectation and we have

$$\begin{aligned}
\mathbb{E}\left[\|z_{t+1} - z_t\|^2\right] &= \left(\frac{\alpha}{1-\beta}\right)^2\mathbb{E}\left[\|m_{t+1} - \beta m_t\|^2\right] \\
&\leq 2\left(\frac{\alpha}{1-\beta}\right)^2\left(\mathbb{E}\left[\|m_{t+1} - (\beta m_t + g_{t+1})\|^2\right] + \mathbb{E}\left[\|g_{t+1}\|^2\right]\right) \\
&\leq 2\left(\frac{\alpha}{1-\beta}\right)^2\left(\mathbb{E}\left[\|g_{t+1}\|^2\right] + \sigma_m^2\right).
\end{aligned}$$

$\square$

*Proof of Theorem 1.* From Lemma 2, we have

$$\mathbb{E}\left[\|z_{t+1} - z_t\|^2\right] \leq 2\left(\frac{\alpha}{1-\beta}\right)^2\left(\mathbb{E}\left[\|g_{t+1}\|^2\right] + \sigma_m^2\right).$$

Substituting Lemma 1 gives

$$\mathbb{E}\left[\|z_{t+1} - z_t\|^2\right] \leq 2\left(\frac{\alpha}{1-\beta}\right)^2\left(\|\nabla f(\theta_t)\|^2 + \sigma^2 + \sigma_m^2\right). \tag{8}$$

Suppose $\theta_*$ is the optimal parameter and $f_* = f(\theta_*)$ is the minimal objective value. First, we have

$$\|z_{t+1} - \theta_*\|^2 = \|z_t - \theta_*\|^2 + 2 \langle z_t - \theta_*, z_{t+1} - z_t \rangle + \|z_{t+1} - z_t\|^2$$

Take expectation over the randomness in the $(t+1)-$th step, we have

$$\mathbb{E}[\|z_{t+1} - \theta_*\|^2] = \|z_t - \theta_*\|^2 - \frac{2\alpha}{1 - \beta} \langle z_t - \theta_*, \nabla f(\theta_t) \rangle + \mathbb{E}[\|z_{t+1} - z_t\|^2]$$

$$= \|z_t - \theta_*\|^2 - \frac{2\alpha}{1 - \beta} \langle \theta_t - \theta_*, \nabla f(\theta_t) \rangle$$

$$- \frac{2\alpha\beta}{(1 - \beta)^2} \langle \theta_t - \theta_{t-1}, \nabla f(\theta_t) \rangle + \mathbb{E}[\|z_{t+1} - z_t\|^2]$$

Since $f$ is continuously differentiable and L-smooth, we have the following inequalities. [34]

$$\langle \theta_t - \theta_*, \nabla f(\theta_t) \rangle \geq \frac{1}{L} \|\nabla f(\theta_t)\|^2 \tag{9}$$

$$\langle \theta_t - \theta_*, \nabla f(\theta_t) \rangle \geq f(\theta_t) - f_* + \frac{1}{2L} \|\nabla f(\theta_t)\|^2 \tag{10}$$

$$\langle \theta_t - \theta_{t-1}, \nabla f(\theta_t) \rangle \geq f(\theta_t) - f(\theta_{t-1}) \tag{11}$$

Substitute them and get

$$\mathbb{E}[\|z_{t+1} - \theta_*\|^2] \leq \|z_t - \theta_*\|^2 - \frac{2\alpha(1 - \rho)}{L(1 - \beta)} \|\nabla f(\theta_t)\|^2 - \frac{2\alpha\rho}{1 - \beta}(f(\theta_t) - f_*)$$

$$- \frac{\alpha\rho}{L(1 - \beta)} \|\nabla f(\theta_t)\|^2 - \frac{2\alpha\beta}{(1 - \beta)^2}(f(\theta_t) - f(\theta_{t-1})) + \mathbb{E}[\|z_{t+1} - z_t\|^2]$$

where $\rho \in (0, 1]$ is a parameter used to balance the first two inequalities. Denote $M = 2\left(\frac{\alpha}{1-\beta}\right)^2 (\sigma^2 + \sigma_m^2)$. Substitute Eq. 8 into this inequality and collect the terms, we get

$$\left(\frac{2\alpha\rho}{1 - \beta} + \frac{2\alpha\beta}{(1 - \beta)^2}\right)(f(\theta_t) - f_*) + \mathbb{E}[\|z_{t+1} - \theta_*\|^2]$$

$$\leq \frac{2\alpha\beta}{(1 - \beta)^2}(f(\theta_{t-1}) - f_*) + \|z_t - \theta_*\|^2 + \left(\frac{2\alpha^2}{(1 - \beta)^2} - \frac{\alpha(2 - \rho)}{L(1 - \beta)}\right)\|\nabla f(\theta_t)\|^2 + M$$

When $\alpha$ satisfies the condition $\frac{2\alpha^2}{(1-\beta)^2} - \frac{\alpha(2-\rho)}{L(1-\beta)} \leq 0$, i.e. $0 \leq \alpha \leq \frac{(1-\beta)(2-\rho)}{2L}$, the term about $\|\nabla f(\theta_t)\|^2$ is non-positive, thus we have

$$\left(\frac{2\alpha\rho}{1 - \beta} + \frac{2\alpha\beta}{(1 - \beta)^2}\right)(f(\theta_t) - f_*) + \mathbb{E}[\|z_{t+1} - \theta_*\|^2]$$

$$\leq \frac{2\alpha\beta}{(1 - \beta)^2}(f(\theta_{t-1}) - f_*) + \|z_t - \theta_*\|^2 + M$$

Summing this inequality from 0 to $T - 1$ and taking full expectation gives

$$\frac{2\alpha\rho}{1 - \beta} \sum_{i=0}^{T-1} \mathbb{E}[f(\theta_i) - f_*] + \sum_{i=0}^{T-1} \left(\frac{2\alpha\beta}{(1 - \beta)^2}\mathbb{E}[f(\theta_i) - f_*] + \mathbb{E}[\|z_{i+1} - \theta_*\|^2]\right)$$

$$\leq \sum_{i=0}^{T-1} \left(\frac{2\alpha\beta}{(1 - \beta)^2}\mathbb{E}[f(\theta_{i-1}) - f_*] + \mathbb{E}[\|z_i - \theta_*\|^2]\right) + T \cdot M$$

which implies that

$$\frac{2\alpha\rho}{1 - \beta} \sum_{i=0}^{T-1} \mathbb{E}[f(\theta_i) - f_*] \leq \frac{2\alpha\beta}{(1 - \beta)^2}(f(\theta_0) - f_*) + \|\theta_0 - \theta_*\|^2 + T \cdot M$$

Since $f$ is *convex*, we have $Tf(\bar{\theta}_T) \leq \frac{1}{T}\sum_{i=0}^{T-1} f(\theta_i))$. Subsequently we have

$$\mathbb{E}[f(\bar{\theta}_T) - f_*] \leq \frac{1}{T}\left(\frac{\beta}{\rho(1-\beta)}(f(\theta_0) - f_*) + \frac{1-\beta}{2\alpha\rho}\|\theta_0 - \theta_*\|^2\right)$$
$$+ \frac{1-\beta}{2\alpha\rho}M$$

Finally, when $\alpha \in (0, \frac{1-\beta}{L}]$, we can take $\rho = 1$, use L-smooth condition again and substitute $M$, which gives

$$\mathbb{E}[f(\bar{\theta}_T) - f_*] \leq \frac{1}{2T}\left(\frac{L\beta}{1-\beta} + \frac{1-\beta}{\alpha}\right)\|\theta_0 - \theta_*\|^2$$
$$+ \frac{\alpha\sigma^2}{(1-\beta)} + \frac{\alpha\sigma_m^2}{(1-\beta)}$$

$\square$

