# OpenReview forum: "Memory Efficient Optimizers with 4-bit States"
_NeurIPS.cc/2023/Conference — NeurIPS 2023 spotlight_

### Official Review · Reviewer_f2sV · 2023-07-06

**Soundness:** 2 fair
**Presentation:** 3 good
**Contribution:** 2 fair
**Rating:** 6
**Confidence:** 3

**Summary:**

This paper concentrates on quantizing the optimizer states into low-bit to reduce training memory consumption. By thoroughly analyzing the first and second momentums, the authors quantize the optimizer states to 4-bit via the use of a smaller block size and both row-wise and column-wise information. They perform extensive experiments across various tasks.

**Strengths:**

(1) The authors thoroughly analyze the outliers of momentums, which makes the existing block-wise method not work well.

(2) They perform extensive experiments across various tasks.

**Weaknesses:**

Compressing the optimizer states is essential especially for training large language models, but the paper only deals with how much memory is saved when fine-tuning LLaMA-7B on Alpaca. It would be necessary to evaluate LLaMA-7B fine-tuned via 4-bit AdamW on common sense reasoning tasks (to check whether the common season reasoning could be preserved even after fine-tuning via 4-bit AdamW) and/or MMLU (to check whether fine-tuned models via 4-bit AdamW can possess the instruction-following ability as a few-shot learner). In addition, it would be much more convincing if the experimental results of LLaMA-13B, 33B, and/or 65B are provided.

**Questions:**

N/A

---

> ### Author Rebuttal · Authors · 2023-08-10
>
> We thank reviewer f2sV for the constructive comments. With respect to your questions:
>
> **Weakness: Common sense reasonging and MMLU evaluations on large models.**
> We have fine-tuned LLaMA-7B and LLaMA-13B with 32-bit AdamW and 4-bit AdamW on Alpaca and evaluated them on common sense reasoning tasks and MMLU. Results show that 4-bit AdamW will not destory the capability of pretraiend models while enabling them to obtain instruction-following ability. In MMLU and HellaSwag tasks, 4-bit AdamW fine-tuning outperforms 32-bit AdamW both on LLaMA-7B and LLaMA-13B. In other tasks, 4-bit AdamW is comparable with 32-bit AdamW.
> 4-bit AdamW does not get worse than 32-bit AdamW when the model size grows.
>
> |    LLaMA-7B | MMLU(5-shot) | HellaSwag | ARC-e | ARC-c | OBQA |
> |-------------|-------------:|----------:|------:|------:|-----:|
> | Original          |          33.1|       73.0|   52.4|   40.9|  42.4|
> | FT w. 32-bit AdamW|          38.7|       74.6|   61.5|   45.1|  43.4|
> | FT w. 4-bit AdamW |          38.9|       74.7|   61.2|   44.4|  43.0|
>
> |    LLaMA-13B| MMLU(5-shot) | HellaSwag | ARC-e | ARC-c | OBQA |
> |-------------|-------------:|----------:|------:|------:|-----:|
> | Original    |          47.4|       76.2|   59.8|   44.5|  42.0|
> | FT w. 32-bit AdamW|          46.5|       78.8|   63.6|   48.3|  45.2|
> | FT w. 4-bit AdamW |          47.4|       79.0|   64.1|   48.0|  45.2|

---

> > ### Comment · Reviewer_f2sV · 2023-08-19
> >
> > Thanks for your clarification. It would be better if the experimental results for LLaMA-33B or LLaMA-65B was given, but most of my concerns are well addressed. However, I have a question. For both LLaMA-7B and LLaMA-13B, FT w. 4-bit AdamW is always better than FT w. 32-bit AdamW on MMLU (5-shot). Could you provide the insight into such unexpected results?

---

> > > ### Author Response · Authors · 2023-08-19
> > > **Second Response to Reviewer f2sV**
> > >
> > > Thank you again for your comments. Regrading your concern:
> > >
> > > **Experiments on LLaMA-33B**
> > >
> > > We fine-tuned LLaMA-33B with both 32-bit AdamW and 4-bit AdamW on Alpaca.
> > > The hyperparameters used are consistent with those of LLaMA-13B.
> > > Notably, both the 32-bit AdamW and 4-bit AdamW achieve comparable training losses of 0.057 and 0.059, respectively. Furthermore, their convergence curves align closely.
> > > Both the 32-bit AdamW and 4-bit AdamW achieve similar training losses of 0.057 and 0.059, respectively. Moreover, their convergence curves closely align.
> > > The results of MMLU and common sense reasoning is reported in the following table.
> > > Unlike LLaMA-7B and LLaMA-13B, on LLaMA-33B, the performance of 4-bit AdamW is lower than 32-bit AdamW on the MMLU task.
> > > Therefore, 4-bit AdamW is not always better than 32-bit AdamW on MMLU task.
> > > However, it's noteworthy that 4-bit AdamW exhibits lower MMLU loss on LLaMA-33B.
> > > Additionally, note that for the 33B model, instruction tuning does not improve performance much, so the implication of the performance might be limited.
> > >
> > > |    LLaMA-33B| MMLU loss (5-shot) |MMLU (5-shot) | HellaSwag | ARC-e | ARC-c | OBQA |
> > > |-------------|-------------:|--------:|------:|------:|------:|-----:|
> > > | Original    |          2.67| 54.9|       79.3|   58.9|   45.1|  42.2|
> > > | FT w. 32-bit AdamW|    0.98| 56.4|       79.2|   62.6|   47.1|  43.8|
> > > | FT w. 4-bit AdamW |    0.95| 54.9|       79.2|   61.6|   46.6|  45.4|

---

> > > > ### Comment · Reviewer_f2sV · 2023-08-20
> > > >
> > > > Thank you for the detailed response. As my questions are completely addressed, I raise my score to acceptance.

---

> > > > > ### Author Response · Authors · 2023-08-21
> > > > > **Thank you for the valuable review**
> > > > >
> > > > > Thank you for the valuable review and raising the score.

---

### Official Review · Reviewer_d65g · 2023-07-06

**Soundness:** 3 good
**Presentation:** 2 fair
**Contribution:** 4 excellent
**Rating:** 7
**Confidence:** 3

**Summary:**

Today’s machine learning pipeline places a heavy load on GPU memory during training, due to the large number of parameters the optimizers have to maintain during training. This paper proposes a method to reduce the internal states of optimizers from their full-width counterparts to 4-bit numbers, while maintaining nearly the same performance across a range of tasks, such as vision and language models.

**Strengths:**

This paper presented some intriguing observations, such as the 'zero-point problem' discussed in relation to second-order momentum, and the first-order term appears to be robust under quantisation. Additionally, this paper examined how quantisation might be combined with factorisation. The results presented covered a variety of language tasks as well as a vision task with relatively advanced models.

**Weaknesses:**

The figures are too small, making it hard to read. It is also unclear to me how the block size was chosen and whether there are theories to support this decision. Additionally, it would be beneficial to demonstrate a wider range of vision tasks to demonstrate the breadth of the proposed approach.

**Questions:**

- How do one generally pick the block size when we have an unseen model? Do we have theories for backing up the block size picking?
- Are there any other ways to demonstrate the convergence (from proof) rather than showing empirical results?
- Regarding the empirical results, more results might be need on 1) generative models (eg. diffusion) and 2) more vision tasks (eg. segmentation or object detection). The core idea is to demonstrate an efficeint optimizer, then the capability of this optimizer should be then tested on a range of different tasks.

**Limitations:**

I do not think this is applicable to this paper.

---

> ### Author Rebuttal · Authors · 2023-08-10
>
> We thank reviewer d65g for the careful review. With respect to the figures, we will use larger ones in a later revision.
> With respect to your questions:
>
> **Weakness + Question 1: How to pick the block size.**
> There are some rules to choose the block size:
>
> - Firstly, quantizing with a smaller block size approximates the tensors better than a larger block size due to the finer granularity of quantization.
> - Secondly, the block size is limited by the memory. A smaller block size would lead to larger memory overhead (e.g., with fp32 scaling constants, a block size of 32 causes ab extra overhead of 1-bit per parameter).
>
> Empirically, we have shown the quantization error of different block sizes via histograms in Appendix C.3 (Effectiveness of Block Size in Block-wise Normalization), which shows that a block size of 128 exactly reconstructs tensors, better than a block size of 2048.
> In general, we consistently choose a small block size of 128 to reduce quantization error and maximize accuracy, while keeping memory overhead under control.
> Actually, the granularity of quantization affected by block size is independent of the model size, and we suspect that a block size of 128 works well for unseen (large) models.
>
> **Question 2: Theoretical analysis about convergence of low-bit optimizers.**
> The convergence of low-bit optimizers can be guaranteed if their fp32 counterparts converge. Here we provide a theorem about the convergence of quantized SGD with momentum (Algorithm 2 in Appendix F) under some assumptions. We believe that the convergence of low-bit AdamW could be inferred from the convergence of AdamW.
>
> We use $f(\theta)$ to denote the objective function, $\beta$ to denote momentum, $Q$ to denote quantizer on first moment and $\alpha$ to denote learning rate.
> With assumptions:
>
> - (Convexity) The objective function is convex and has an unique global minimum $f(\theta^*)$.
> - (Smoothness) The objective $f(\theta)$ is continuous differentiable and $L$-smooth;
> - (Unbiasedness and Bounded variance) $\mathbb{E}[\nabla\hat{f}(\theta)] = \nabla f(\theta)$ and $ \mathbb{E}\left[\left\lVert\nabla\hat{f}(\theta) - \nabla f(\theta)\right\rVert^2\right] < \sigma^2$, $\forall \theta \in \mathbb{R}^d$, where $\nabla\hat{f}$ is stochastic gradient.
> - (Unbiased quantizer)  $\forall x\in \mathbb R^d$, $\mathbb{E}\left[Q(x)\right]=x$.
> - (Bounded quantization variance) $\forall x\in \mathbb R^d$, $\mathbb{E}\left[\left\lVert Q(x)-x\right\rVert^2\right]\le \sigma_m^2$.
>
> We have the following theorem:
>
> **Theorem 1.** Suppose above assumptions hold.
> Consider the case where quantized SGD with momentum is performed on the objective function $f$, with momentum parameter $\beta$.
> Let $\alpha \in (0, \frac{1-\beta}{3L}]$, then for all $T > 0$ we have
>
> $$
>     \mathbb{E}[f(\bar{\theta_T}) - f_*] \le \frac{1}{2T}\left( \frac{L\beta}{1-\beta} + \frac{1-\beta}{\alpha } \right) \left\lVert\theta_0 - \theta_*\right\rVert^2
>     + \frac{\alpha \sigma^2}{(1-\beta)}
>     + \frac{\alpha\sigma_m^2}{2(1-\beta)}
> $$
>
> where $\bar{\theta_T} = \frac{1}{T}\sum_{i=0}^{T-1}{\theta}_{i}$.
>
> The theorem indicates that when $\alpha\rightarrow 0$ and $\alpha T\rightarrow \infty$, the optimizer converge to a stationary-point.
> Note that the unbiased quantizer assumption is only for technical simplification.
> Our proof mainly follows the prior analysis of SGD with momentum [1, 2].
>
> [1] Ghadimi, Euhanna, Hamid Reza Feyzmahdavian, and Mikael Johansson. "Global convergence of the heavy-ball method for convex optimization." 2015 European control conference (ECC). IEEE, 2015.
>
> [2] Liu, Yanli, Yuan Gao, and Wotao Yin. "An improved analysis of stochastic gradient descent with momentum." Advances in Neural Information Processing Systems 33 (2020): 18261-18271.
>
> **Weakness + Question 3: Results on generative modeling and more vision tasks.** We are running our optimizers on generative modeling and more vision tasks. We will post updates on these tasks within the discussion period.

---

> > ### Author Response · Authors · 2023-08-13
> > **Response to Reviewer d65g (cont.)**
> >
> > **Results on generative modeling and more vision tasks.**
> >
> > We tested our optimizers on generative modeling task and more vision tasks. For vision tasks, we conducted experiments on image classification with ResNet-50 on ImageNet, detection with Faster R-CNN (with ResNet-50 backbone) on COCO, segmentation with Mask R-CNN (with ResNet-50 backbone) on COCO.
> > The optimizers are 4-bit SGDM / 6-bit SGDM, with the first moment compressed to lower precision using our method. These tasks require 6-bit SGDM to achieve lossless performance compared with 32-bit SGDM. Note that the memory consumption of 6-bit SGDM is still lower than 4-bit Adam, as the former only retains the first moment.
> >
> > For generative modeling task, we trained a DDPM++ model on class-conditional CIFAR-10. The results indicate that it is still challenging to pretrain diffusion models with compressed optimizers. We suspect it is because the high variance of the denoising score matching objective. Nevertheless, our proposed optimizer outperforms the existing compressed optimizers proposed by Dettmers et al. Specifically, while Dettmers et al.'s 8-bit Adam diverges on this task, our 8-bit Adam successfully converges and generates meaningful images.
> >
> > |    Method         | ResNet-50 CLS| Faster R-CNN| Mask R-CNN |
> > |-------------------|-------------:|----------:|------:|
> > | 32-bit SGDM       |          77.2|       37.2|   34.3|
> > | 8-bit SGDM (Dettmers et al., 2021)       |          77.2|       N/A |   N/A |
> > | 4-bit SGDM (ours) |          76.7|       33.8|   31.8|
> > | 6-bit SGDM (ours) |          77.3|       37.2|   34.2|
> >
> >
> > |    Method         | FID          |
> > |-------------------|-------------:|
> > | 32-bit Adam       |          1.89|
> > | 8-bit  Adam (Dettmers et al., 2021)      |        655.12|
> > | 4-bit  Adam (ours)|        402.94|
> > | 8-bit  Adam (ours)|         53.56|
> >
> > [1] Dettmers, Tim, et al. "8-bit Optimizers via Block-wise Quantization." International Conference on Learning Representations. 2021.

---

> > ### Comment · Reviewer_d65g · 2023-08-15
> >
> > Thanks for your reply.
> >
> > I may have to clarify my questions on the theory side. It seems like the quantization variance is associated with your block size B, and I am interested in understanding how or would this would affect the convergence speed? If I pick different B values, surely the variance bound provided should be different?
> >
> > I look forward seeing your results on other generative tasks.

---

> > > ### Author Response · Authors · 2023-08-18
> > > **Second Response to Reviewer d65g**
> > >
> > > Thank you again for your comments. Regrading the questions:
> > >
> > > **Results on generative modeling task.**
> > > We trained a StyleGAN2-ADA with both 32-bit Adam and our 4-bit Adam on the conditional CIFAR-10 dataset.
> > > We used the same hyperparameters as the official implementation, except for a larger batch size that facilitated quicker training. The following table shows the best achieved FID score throughout the training process.
> > > During the training dynamics, several quantites, including generator loss, descriminator loss, fake image scores, and real image score closely align for both optimizers.
> > > When evaluated in terms of FID, the two models trained with different optimizers exhibit a slight gap in performance.
> > > Furthermore,
> > > in the earlier conducted diffusion model task, we observed a close alignment in the loss dynamics between the 32-bit Adam and our 8-bit Adam optimizers.
> > >
> > > |    Method         | FID          |
> > > |-------------------|-------------:|
> > > | 32-bit Adam       |          2.40|
> > > | 4-bit  Adam (ours)|          2.89|
> > >
> > > **Theory about block size.**
> > > In our analysis, the quantization variance only impacts the magnitude of the stationary point, i.e., the case when $T \to \infty$, and it does not relate to the convergence speed.
> > > This is similar to the variance from stochastic gradient.
> > > Then, we aim to characterize the influence of block-wise normalization on quantization variance.
> > > When employing a linear quantizer with an interval of $\delta$ (without block-wise normalization), the quantization variance is explicitly given by
> > > $$
> > > \sigma_m^2 = \mathbb{E}\left[||Q_{\delta}(x) - x||^2\right]
> > > \le \frac{\delta^2 d}{4},  \forall x\in \mathbb R^d.
> > > $$
> > > The value of $\delta$ is determined by the maximum absolute value (aka absmax) within the vector and the number of representable values, i.e., the number of bits used.
> > > Upon incorporating block-wise normalization with a block size of $B$, assuming there are $N$ blocks in total, and denoting the interval within each block as $\delta_i$, the quantization variance is given by
> > > $$
> > > \sigma_m^{2} = \mathbb{E}\left[||Q_{\delta}(x) - x||^2\right]
> > > \le \sum_{i=1}^N \frac{\delta_i^2 B}{4},  \forall x\in \mathbb R^d.
> > > $$
> > > However, the extent of improvement in quantization variance through blocking heavily relies on the tensor's structure. Here, we qualitatively analyze the impact of block size in various scenarios:
> > > - In an extreme case, if outliers comparable to, or same as, the absmax value occur within each block, block-wise quantization fails to yield any improvement.
> > > This instance finds empirical support in our investigations, where we have empirically observed regular distribution of outliers within moment tensors along rows and/or columns. This pattern suggests the advantages of employing smaller block sizes.
> > > - When the first moments are i.i.d. from $N(0, 1)$, we have $\delta_i \propto \sqrt{\log(B)}$ thanks to the nice property of the Gaussian distribution.
> > > Consequently, if two distinct block sizes, $B_1$ and $B_2$, are employed in quantization, the quantization variance ratio is given by
> > >     $$
> > >     \frac{\sigma_{m, B_1}^{2}}{\sigma_{m, B_2}^{2}}
> > >     = \frac{\log(B_1)}{\log(B_2)}.
> > >     $$
> > >     For neural network training, considering $B_1 = 128$ and $B_2$ as the size of a single tensor, significant enhancements in quantization variance are achieved. Furthermore, with $B_1 = 128$ and $B_2 = 2048$, the quantization variance improves by 7/11.
> > > - In practical scenarios, the distribution of moments is difficult to characterize analytically, thus making it difficult to determine the impact of block size on quantization variance. We leave the theoretical results for future work and instead present empirical analyses regarding block size and quantization variance. We utilize block-wise normalization with different block sizes to quantize the first moment tensors in a GPT-2 medium model.
> > > We report both the mean and maximum relative quantization errors across tensors. The relative quantization error is defined as $||Q(x) - x||/||x||$.
> > > In this specific setting, the results show a rough log-correlation between relative error and block size.
> > >     |  Block size   | 128 | 256 | 512 | 1024 | 2048|
> > >     |---------------|----:|----:|----:|-----:|-----:|
> > >     | first moment(mean) | 0.158|0.170|0.183 | 0.194|0.205 |
> > >     | first moment(max) | 0.173|0.190|0.208 | 0.222|0.235 |
> > >
> > >     However, it is important to note that the actual performance metric (e.g., accuracy) exhibits a complex relationship with quantization variance, and even training loss, exceeding the scope of our block size analysis.

---

> > > > ### Comment · Reviewer_d65g · 2023-08-18
> > > > **Thanks for your experiments**
> > > >
> > > > I bumped my score to an accept, thanks for taking your time to conduct all the experiments.
> > > > side note: it would be great if you can make the fonts in your figures a bit bigger, eg. from figure 1-3.
> > > >
> > > > Regarding reviewer qs7T, it is important to note that, only the Alpaca LLaMA7B is widely recognized as a STOA result atm, along with possibly the OPT results (before rebuttal). In my initial review, I did not make any mention of the necessity for a grid search for block size, so I am uncertain about the origin of your comments in that regard. The attitude of "whatever works works" is also nonsense. Over the past few years, I have worked on both training and inference quantization and has a track record of publishing both on tier-1 conferences. While it is definitely okay for you to champion a paper, it is unprofessional to assume that you are the sole individual who comprehends its contents and to dismiss the reviewing process. Despite the paper being of high quality, I cannot agree with any words in your general response.

---

> > > > > ### Author Response · Authors · 2023-08-19
> > > > > **Thank you for the valuable review**
> > > > >
> > > > > Thank you for the valuable review and raising the score.
> > > > > We will adjust the font size in the figures in subsequent revision.

---

### Official Review · Reviewer_qs7T · 2023-07-07

**Soundness:** 4 excellent
**Presentation:** 4 excellent
**Contribution:** 4 excellent
**Rating:** 8
**Confidence:** 5

**Summary:**

The paper develops 4-bit optimizers by using a smaller blocksize for the first moment and by analyzing and finding solutions to outlier patterns in the second moment. In particular it is found that quantization to the zeropoint are problematic for the second moment. The analysis is extensive and robust leading to novel and valuable insights into how optimizer states work in the low-bit regime.


Recommendation:

This is a high-quality paper with extensive and valuable analysis and new methods with very robust evaluations. I am happy to fight for acceptance of this paper. I recommend that this paper be highlighted at the conference.

**Strengths:**

- The analysis in this paper is very robust, both theoretically and empirically. In particular, the identification of the zero-point problem is important.
- Rank-1 normalization for quantization is ingenious and a valuable solution that goes beyond blocking/grouping that is done in many other quantization papers.
- This work will have a strong impact on the general quantization literature way beyond quantized optimizers.


**Weaknesses:**

- This is outstanding work. I cannot find any weakness.


**Questions:**

- How do you store Rank-1 normalization constants to perform the dequantization? As I understand it, you use min(absmax row, absmax col) as the normalization constant for each x_{i, j}, but in this case, you need to store which constant is valid for each index (i, j), is that correct?

Suggestion: You often write 2nd momentum. I believe the correct terminology would be first and second moment (look for central moment on Wikipedia). It is correct that the first moment is equivalent to the momentum term in Adam, but I think the second moment is usually not referred to as second momentum (rather a RMSProp term).

**Limitations:**

Limitations are fully and honestly discussed.

---

> ### Author Rebuttal · Authors · 2023-08-10
>
> We thank reviewer qs7T for the valuable and careful review.
>
> Consider a weight matrix $x$ of size $m \times n$. In Rank-1 normalization, we store the absmax of each row and the absmax of each column during the quantization stage, totaling $m + n$ elements.
> In the dequantization stage, we calculate the normalization constant for each entry in the same manner as in the quantization stage, i.e., min(absmax row_i, absmax col_j) for entry $x_{i, j}$.
> The subtle difference is that we do not need to calculate the absmax of the i-th row (and the j-th colum) but use the pre-stored values instead.
> Additionally, thank you for pointing out the typos concerning the first/second moment. We will correct them in a later revision.

---

### Official Review · Reviewer_rPDD · 2023-07-07

**Soundness:** 3 good
**Presentation:** 3 good
**Contribution:** 3 good
**Rating:** 5
**Confidence:** 5

**Summary:**

This paper proposes a 4-bit optimizer to save the memory of model training. They use a smaller block size and propose to utilize both row-wise and column-wise information for better quantization. They identify a zero-point problem of quantizing the second-order momentum and solve this problem with a linear quantizer that excludes the zero point.

**Strengths:**

1. aims at solving a key problem to reduce the memory occupation for model training
2. conduct experiments on vision and language tasks

**Weaknesses:**

1. lack experiments on large models like 65B or 175B
2. The accuracy achieved by the 4-bit optimizer is still lower than the fp16 or int8 version. It is unsure whether it can behave stably under various settings.

**Questions:**

1. What about the effect on large models
2. Comparison with other optimization methods for large models (e.g., https://arxiv.org/abs/2306.09782)

**Limitations:**

This method can not achieve lossless accuracy.

The stability is not clear and sufficiently validated.

---

> ### Author Rebuttal · Authors · 2023-08-10
>
> We thank reviewer rPDD for the constructive review. With respect to your questions:
>
> **Weakness 1 + Question 1: Experiments on large models.**
> We have fine-tuned LLaMA-7B and LLaMA-13B with 32-bit AdamW and 4-bit AdamW on Alpaca and evaluated them on common sense reasoning tasks and MMLU. Results show that 4-bit AdamW will not destory the capability of pretraiend models while enabling them to obtain instruction-following ability. In MMLU and HellaSwag tasks, 4-bit AdamW fine-tuning outperforms 32-bit AdamW both on LLaMA-7B and LLaMA-13B. In other tasks, 4-bit AdamW is comparable with 32-bit AdamW.
> 4-bit AdamW does not get worse than 32-bit AdamW when the model size grows. Also, we are trying to fine-tune LLaMA-30B, and will post updates when available. However, the amount of computation required for LLaMA-65B exceeds our accessible resources.
>
> |    LLaMA-7B | MMLU(5-shot) | HellaSwag | ARC-e | ARC-c | OBQA |
> |-------------|-------------:|----------:|------:|------:|-----:|
> | Original          |          33.1|       73.0|   52.4|   40.9|  42.4|
> | FT w. 32-bit AdamW|          38.7|       74.6|   61.5|   45.1|  43.4|
> | FT w. 4-bit AdamW |          38.9|       74.7|   61.2|   44.4|  43.0|
>
> |    LLaMA-13B| MMLU(5-shot) | HellaSwag | ARC-e | ARC-c | OBQA |
> |-------------|-------------:|----------:|------:|------:|-----:|
> | Original    |          47.4|       76.2|   59.8|   44.5|  42.0|
> | FT w. 32-bit AdamW|          46.5|       78.8|   63.6|   48.3|  45.2|
> | FT w. 4-bit AdamW |          47.4|       79.0|   64.1|   48.0|  45.2|
>
> **Weakness 2: About lossless accuracy and results on more settings.**
> It is true that our 4-bit optimizer does not converge losslessly on all tasks. However, our 4-bit optimizer can achieve lossless results on all language model fine-tuning tasks, including fine-tuning RoBERTa-Large on GLUE and SQuAD, fine-tuning GPT-2 Medium on E2E-NLG, and fine-tuning LLaMA-7B and LLaMA-13B on Alpaca. This already demonstrates its practical applicability.
>
> We are also running our optimizers on other tasks. We will post updates on these tasks within the discussion period.
>
> **Question 2: Comparison with other optimization methods for large models.**
> Recently, there have been many works focusing on optimization for LLMs.
>
> Sophia [1] is a second-order optimizer designed for language model pretraining.
> Sophia converges faster than the broadly used AdamW on GPT-2 pretraining task,
> *but it does not concern about memory efficiency and it has same memory cost as AdamW*.
>
> LOMO [2] is a memory efficient optimizer for fine-tuning language models.
> LOMO is essentially a vanilla SGD optimizer without momentum, and thus it naturally removes the memory consumption of optimizer states. Then, LOMO fuses backpropagation and optimizer update into one step to further reduce memory consumption of gradients and full-precision copies of weights. *However, as LOMO is just a vanilla SGD optimizer, its performance may be worse than Adam in some tasks.*
> Instead, our 4-bit optimizers focus on reducing memory usage of the first and second moments in stateful optimizers (e.g., Adam), and has a wider range of applicability.
> The fusion technique in LOMO and our 4-bit optimizers complement each other, and could bring a more memory efficient Adam while maintaining performance.
>
> MeZO [3] is a memory efficient zeroth-order optimizer for fine-tuning language models. MeZO adapts the ZO-SGD algorithm by using in-place operation and resetting random seed. It only requires the same memory as inference (i.e., only forward parameters).
> *However, as MeZO does not utilize gradient, its performance is worse than gradient-based optimizers.*
> Compared with MeZO, our 4-bit AdamW achieves better results on language model fine-tuning (including fine-tuning RoBERTa-Large on GLUE and SQuAD, GPT-2 Medium on E2E-NLG, and LLaMA-7B and LLaMA-13B on Alpaca). On the other hand, MeZO has performance gaps in some tasks compared with full parameter fine-tuning with Adam and it needs more steps to achieve strong performance.
>
> In summary, LOMO and MeZO are designed for fine-tuning language models with strong simplifications of the optimizer (vanilla SGD / zeroth-order optimization) and *they may not be applicable for pretraining.* In contrast, our 4-bit AdamW is essentially an AdamW, and should have similar range of applicability with AdamW.
>
> Finally, we notice that [1-3] are all contemporaneous work with our submission.
>
> [1] Liu, Hong, et al. "Sophia: A Scalable Stochastic Second-order Optimizer for Language Model Pre-training." arXiv preprint arXiv:2305.14342 (2023).
>
> [2] Lv, Kai, et al. "Full Parameter Fine-tuning for Large Language Models with Limited Resources." arXiv preprint arXiv:2306.09782 (2023).
>
> [3] Malladi, Sadhika, et al. "Fine-Tuning Language Models with Just Forward Passes." arXiv preprint arXiv:2305.17333 (2023).

---

> > ### Author Response · Authors · 2023-08-13
> > **Response to Reviewer rPDD (cont.)**
> >
> > **Results on more settings**
> >
> > We tested our optimizers on generative modeling task and more vision tasks. For vision tasks, we conducted experiments on image classification with ResNet-50 on ImageNet, detection with Faster R-CNN (with ResNet-50 backbone) on COCO, segmentation with Mask R-CNN (with ResNet-50 backbone) on COCO.
> > The optimizers are 4-bit SGDM / 6-bit SGDM, with the first moment compressed to lower precision using our method. These tasks require 6-bit SGDM to achieve lossless performance compared with 32-bit SGDM. Note that the memory consumption of 6-bit SGDM is still lower than 4-bit Adam, as the former only retains the first moment.
> >
> > For generative modeling task, we trained a DDPM++ model on class-conditional CIFAR-10. The results indicate that it is still challenging to pretrain diffusion models with compressed optimizers. We suspect it is because the high variance of the denoising score matching objective. Nevertheless, our proposed optimizer outperforms the existing compressed optimizers proposed by Dettmers et al. Specifically, while Dettmers et al.'s 8-bit Adam diverges on this task, our 8-bit Adam successfully converges and generates meaningful images.
> >
> > |    Method         | ResNet-50 CLS| Faster R-CNN| Mask R-CNN |
> > |-------------------|-------------:|----------:|------:|
> > | 32-bit SGDM       |          77.2|       37.2|   34.3|
> > | 8-bit SGDM (Dettmers et al., 2021)       |          77.2|       N/A |   N/A |
> > | 4-bit SGDM (ours) |          76.7|       33.8|   31.8|
> > | 6-bit SGDM (ours) |          77.3|       37.2|   34.2|
> >
> >
> > |    Method         | FID          |
> > |-------------------|-------------:|
> > | 32-bit Adam       |          1.89|
> > | 8-bit  Adam (Dettmers et al., 2021)      |        655.12|
> > | 4-bit  Adam (ours)|        402.94|
> > | 8-bit  Adam (ours)|         53.56|
> >
> > [1] Dettmers, Tim, et al. "8-bit Optimizers via Block-wise Quantization." International Conference on Learning Representations. 2021.
> >
> > **Experiments on LLaMA-30B**
> >
> > We finetuned LLaMA-30B with 32-bit AdamW and 4-bit AdamW on Alpaca and evaluated them on common sense reasoning tasks and MMLU. The hyperparameters used are consistent with those of LLaMA-13B. Both the 32-bit AdamW and 4-bit AdamW achieve similar training losses of 0.057 and 0.059, respectively. Moreover, their convergence curves closely align. The few-shot / zero-shot performance is reported in the following table.
> > In the case of MMLU, while the accuracy of 4-bit AdamW is lower compared with 32-bit AdamW, it does exhibit a lower loss value.
> > Note that for the 30B model, instruction tuning does not improve performance much, so the implication of the performance might be limited.
> >
> > |    LLaMA-30B| MMLU loss (5-shot) 	 |MMLU (5-shot) | HellaSwag | ARC-e | ARC-c | OBQA |
> > |-------------|-------------:|------------:|------:|------:|-----:|-----:|
> > | Original    |          2.67|54.9|       79.3|   58.9|   45.1|  42.2|
> > | FT w. 32-bit AdamW|    0.98| 56.4|       79.2|   62.6|   47.1|  43.8|
> > | FT w. 4-bit AdamW |    0.95| 54.9|       79.2|   61.6|   46.6|  45.4|

---

> > ### Comment · Reviewer_rPDD · 2023-08-20
> >
> > Thanks for your response. In Table 3, you give the time comparison of 32-bit AdamW, 8-bit AdamW, and 4-bit AdamW. It would be better to give a detailed presentation of the extra time overhead brought by the compress and decompress operation. Further, the time comparison may also become different since the weight size is larger and the elementwise compression might be heavier.

---

> > > ### Author Response · Authors · 2023-08-21
> > > **Second Response to Reviewer rPDD**
> > >
> > > Thanks again for the comments. Regarding your concern:
> > >
> > > **Extra time overhead brought by compression.**
> > > We tested the time overhead introduced by the quantization of our 4-bit AdamW on all tasks listed in Table 3. The results are displayed in the following table. Dettmers et al.'s 8-bit optimizers fuse quantization and optimizer update into a single operator, making it difficult to measure the separate time usage of quantization. We also offer a fused implementation of our 4-bit AdamW and give its time and memory usage results.
> > >
> > > - For 4-bit AdamW without operator fusion, RoBERTa-Large and GPT-2 Medium, which have similar sizes, exhibit similar ratios of quantization time overhead over total time, at 32% and 26%, respectively. For LLaMA-7B, the quantization time overhead is exceedingly low compared to the total time. This is due to our use of a large gradient accumulation step of 32, and the fact that the communication overhead is more significant than the computation. More generally, in the FSDP or ZeRO settings, the time overhead resulting from quantization is determined by the size of the parameter slice on a single node, as 4-bit AdamW would quantize the optimizer state slices independently and in parallel. The speedup of 4-bit AdamW compared to 32-bit AdamW may stem from larger memory buffers/cache due to memory saving of the optimizer states.
> > > - 4-bit AdamW (fused) is always faster than 32-bit AdamW. On RoBERTa-Large and GPT-2 Medium, the speedup of 4-bit AdamW (fused) is due to faster optimizer operation. On LLaMA-7B, the speedup of 4-bit AdamW (fused) is due to the reduced global memory footprint (low-bit read/write and/or large memory cache).
> > >
> > >
> > > |    Task         | Optimizer    | Total Time|  Quantization Time Overhead | Total Mem. |
> > > |-----------------|-------------|----------:|---:|---:|
> > > | LLaMA-7B        |  32-bit AdamW |  3.35 h |  N/A | 75.40 GB |
> > > | LLaMA-7B        |  4-bit AdamW  |  3.07 h |  14 s| 31.87 GB |
> > > | LLaMA-7B        |  4-bit AdamW (fused)|  3.11 h| N/A| 31.88 GB |
> > > | RoBERTa-Large   |  32-bit AdamW |  3.93 min |  N/A | 5.31 GB  |
> > > | RoBERTa-Large   |  4-bit AdamW| 5.59 min | 1.80 min| 3.02 GB |
> > > | RoBERTa-Large   |  4-bit AdamW (fused)|   3.17 min|   N/A| 3.00 GB |
> > > | GPT-2 Medium    |  32-bit AdamW |  2.13 h |  N/A | 6.89 GB |
> > > | GPT-2 Medium    |   4-bit AdamW |  2.43 h | 0.63 h| 4.62 GB |
> > > | GPT-2 Medium    |   4-bit AdamW (fused)| 2.11 h|  N/A | 4.62 GB |

---

### Comment · Reviewer_qs7T · 2023-08-16
**I want to champion this paper**

This is an outstanding paper that I want to see accepted, and I feel some other reviewers lack the expertise to evaluate this work. I worked with low-bit optimizers before and have significant experience of how experimental results and methods should be evaluated.

If this paper does not get accepted, I will not review for the next NeurIPS conference.

I will rebut the other reviewer's responses in turn.

rPDD:

- The concern for lack of large scale is unreasonable. The authors have a very extensive experimental setup and training larger models will require more than 10,000 GPU hours per training run. This is not sensible. Additionally, the machine learning baseline is a very difficult baseline to do well on. The method introduced by the authors does well on this. These results indicate that this optimizer scales well.
- Comparison with the work https://arxiv.org/abs/2306.09782 is not relevant since it deals with a completely different research direction. It is orthogonal. I believe this work my the work of the reviewer who wants a citation

d65g
- More experimental results. Again this is unreasonable. The experimental setup by the authors is more robust than 99% of papers you will see at NeurIPS. They have computer vision, language, natural language understanding, instruction tuning, zero-shot LLM modeling, in-context learning, machine translation. And you request more experiments?
- I think the theoretical justification of the method will not improve the paper. We have so many theoretic justifications for optimizers and no optimizer is better than AdamW. I think this paper does outstanding empirical work, and we need more of this.
- justification for blocksize: Its impossible to run grid searches of blocksize selections across experiments. The authors use a very reasonable blocksize based on the empirical data that is available in the literature. All of this is very sound and good science to base on previously established methods.

 f2sV
- The requested fine-tuning experiments were done with outstanding results. I think this reviewer just forgot to increase their score.

---

> ### Author Response · Authors · 2023-08-18
> **Thank you for the comments!**
>
> A heartfelt thank you goes out for your supportive comments and your enthusiasm regarding our work!
> We also deeply appreciate the constructive feedback from all the reviewers. We believe that through the insights of all reviewers, we can further improve the quality of our work.
> Thanks again for your support!

---

### Decision · Program_Chairs · 2023-09-21

**Decision:**

Accept (spotlight)

**Comment:**

Solid paper on training with "truly" 4-bit optimizers. Most existing work on low-precision training still uses higher-prevision accumulators; this paper proposes a recipe to bypass that limitation. The reviewers all agreed on acceptance, and I think this work will be of great interest to the community.